# Acoustic modulation of mechanosensitive genes and adipocyte differentiation
Masahiro Kumeta [1,2] ✉, Makoto Otani[3], Masahiro Toyoda[4] & Shige H. Yoshimura [1,2]

Eukaryotic cells are equipped with multiple mechanosensory systems and perceive a wide range of mechanical stimuli from the environment. However, cell-level responses to audible range of acoustic waves, which transmit feeble yet highly frequent physical perturbations, remain largely unexplored. Here, we established a direct sound emission system with a vibrational transducer, and acoustic waves at frequency 440 Hz, 14 kHz, and white noise were transmitted to the murine C2C12 myoblasts at 100 Pa intensity. After 2 and 24 h sound emission, 42 and 145 differentially expressed genes, respectively, were identified using RNA-sequencing. Both cell- and sound-related factors important for inducing gene responses were further investigated. The activation of prostaglandin-endoperoxide synthase 2/cyclooxygenase-2 (*Ptgs2/Cox-2*), a high and immediate sound-responding gene, is dependent on focal adhesion kinase activation and mediates sound-triggered gene responses by activating prostaglandin E2 synthesis. Adipocyte cells exhibited prominently high sound responses, and their differentiation was significantly suppressed by continuous or periodic acoustic stimulation. Collectively, these findings redefine acoustic waves as cellular stimulators and provide new avenues for applying acoustic techniques in biosciences.

Eukaryotic and prokaryotic cells are equipped with various mechanosensory systems, typically comprising mechanically gated channels, membrane-linked signalling systems, and cytoskeleton-associated molecular sensors. Forces ranging from several picoNewton to nanoNewton are directly sensed by this molecular machinery to induce adaptive cellular responses[1]. Studies focusing on cell-level responses to contact pressure, osmotic pressure, shear stress, stretch and compression forces, and substrate elasticity have revealed cellular strategies that utilise mechanical stimuli as vital external information to optimise their internal activities[1–4]. Such mechanical stimuli are essential for maintaining mechanosensitive tissues, such as bone, muscle, and adipose tissues, where they are used by cells to regulate proliferation, differentiation, metabolic activity, and cell death.

Sound, one of the most ubiquitous physical forces in nature, is a compressional mechanical wave that transmits oscillating and fluctuating pressure through substances. In a standard physiological environment, the audible range, which has a frequency range of approximately 20 Hz to 20 kHz, transmits several pascals (Pa) of pressure in air and several kPa in water. In addition to air-conducted sound sensed by the hearing system, two major types of sound transmission (bone- and soft-tissue-conducted sound) form a complicated acoustic environment inside the body[5]. Instead of the classically reported 5 N static force, which is the threshold

for activating bone conduction, soft-tissue conduction was recently shown to work at a 0 N application force without direct physical contact between the sound source and body[6,7]. Research on sheep that measured the transmission of external audible sound to the foetus inside the uterus revealed approximately 5−7 decibel (dB) attenuation: 106–107 dB output of white noise (broadband noise) from a loudspeaker transmitted approximately 2 Pa pressure deep inside the body[8,9]. Meanwhile, physical contact causes a much higher body sound transmission than external audible sound. More specifically, when a mechanical force equivalent to normal human exercise was applied to an elastic object mimicking soft biological tissues, the simulated pressure was ~400 kPa (Supplementary Fig. 1). Both simulation and experimental studies have revealed the ability of hard and soft body tissues to transmit several kPa of pressure distally by several centimetres[10,11], suggesting that, under physiological conditions, the sound pressure transmitted to cells in living tissues is several kPa in magnitude. However, sound and acoustic waves have not been extensively investigated as sources of cellular stimulation. Here, in this study, we investigate how cells respond to the physiological range of acoustic irradiation that defines the biological significance of sound as a mechanical stimulation and uncover the fundamental relationships between life and sound.

[1]Graduate School of Biostudies, Kyoto University, Kyoto, Japan. [2]Center for Living Systems Information Science (CeLiSIS), Kyoto University, Kyoto, Japan. [3]Graduate School of Engineering, Kyoto University, Kyoto, Japan. [4]Faculty of Environmental and Urban Engineering, Kansai University, Osaka, Japan. ✉e-mail: kumeta@lif.kyoto-u.ac.jp; kumeta.masahiro.4r@kyoto-u.ac.jp

## Results

### Identification of genes responsive to acoustic stimulation

We have previously established a sound emission system using a conventional loudspeaker that transmits several mPa of acoustic pressure, altering mechanosensitive genes[12]. In this study, we established a direct sound emission system using a vibrational transducer to generate acoustic waves directly in the culture medium (Fig. 1a, Supplementary Fig. 2a–c). A custom-made vibrating plate made of polyether ether ketone (PEEK) plastic was selected owing to its high mechanical strength, low weight, and low heat conductivity (Supplementary Fig. 2d–f). A set of sound patterns, including single-frequency sound and white noise, was generated using the NCH Tone Generator software (Supplementary Audio 1–5). The sound intensity was measured directly by recording sound in water using a hydrophone, and the pressure level was calculated. Using this system, a variety of acoustic waves can directly be transmitted to cultured cells at the maximum intensity of approximately 100 Pa, without transmitting heat from the transducer

(Supplementary Fig. 2f), which is within the physiological range of sound pressure in living tissues.

The murine C2C12 myoblast cell line was selected to investigate acoustic responses since it showed significant gene responses in our previous study[12]. The cells were cultured in plastic dishes at approximately 50% confluence and subjected to acoustic stimulation. Single-frequency sine-wave sounds at 440 Hz and 14 kHz were selected as representatives of low and high audible frequencies, respectively, while white noise was selected as a broadband source with uniform frequency characteristics. After 2 h and 24 h of continuous sound emission at 100 Pa, total RNA was extracted and subjected to gene expression profiling using RNA-sequencing. In total, 42 early- and 145 late-response genes were identified as sound-sensitive after 2 h and 24 h of stimulation, respectively (Fig. 1b, c). The 42 early-response genes included 33 upregulated and 9 downregulated genes. Among these, five exhibited consistent upregulation in all three sound patterns. The 145 late-response genes included 86 upregulated, 52 downregulated, and 7 dual-

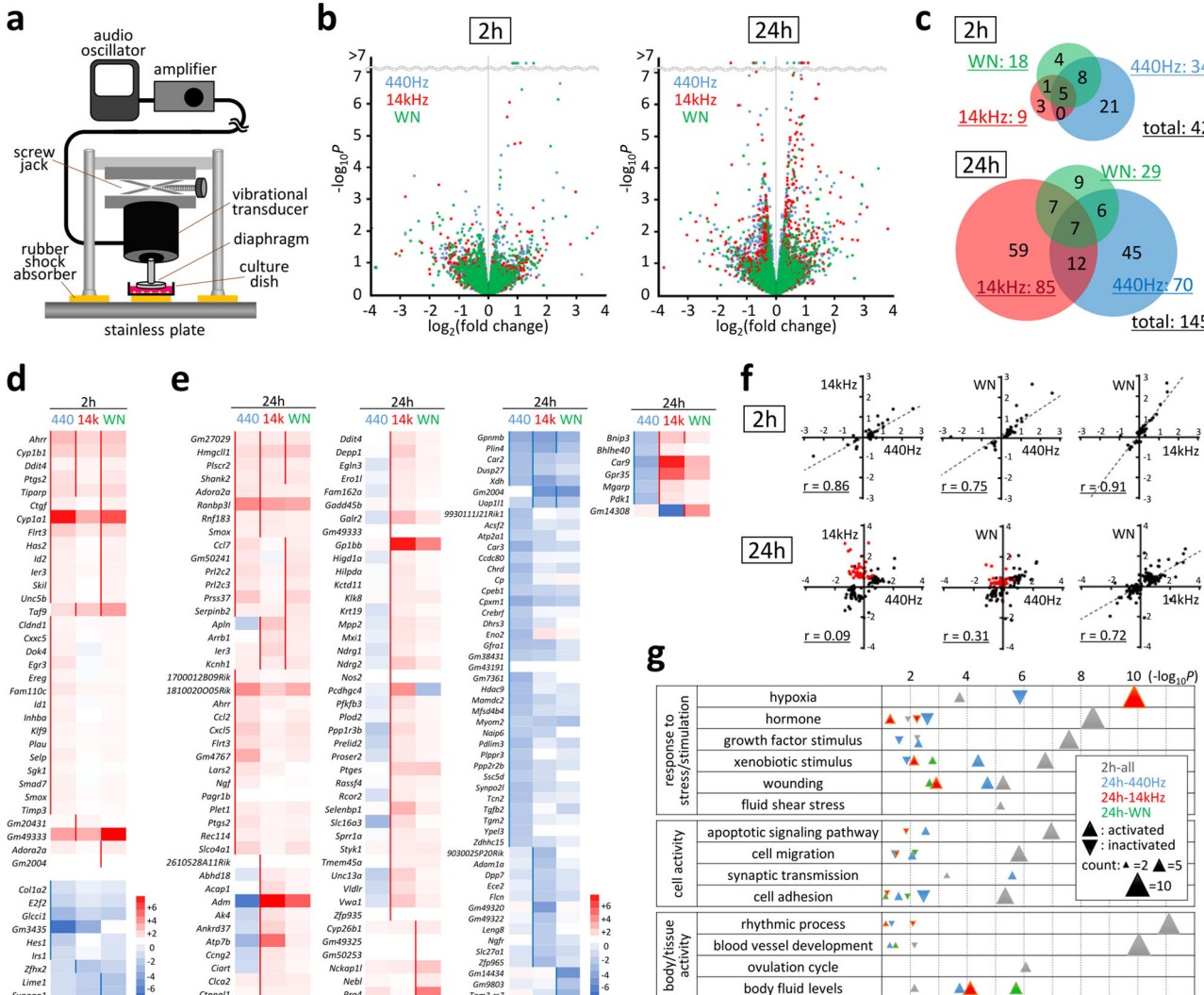

**Fig. 1 | Identification of sound-sensitive genes. a** Schematic illustration of the direct sound emission system established in this study. The detailed setting procedure and monitoring/quantifying methods are described in Supplementary Fig. 2. **b** Volcano plots showing differentially expressed genes in C2C12 cells after 2 h and 24 h of continuous emission of 440 Hz (blue), 14 kHz (red) and white noise (green) sound at 100 Pa, identified using RNA-sequencing analyses. Samples were analysed in triplicate for each condition. **c** Venn diagrams of sound-sensitive genes identified using RNA-sequencing analyses. In total, 42 and 145 genes were identified after 2 h and 24 h of continuous emission of 440 Hz (blue), 14 kHz (red) and white noise

(green) sound, respectively, at 100 Pa. **d** Heat map for 33 activated and 9 inactivated genes after 2 h of sound emission. Positive and negative fold changes are shown on a red-to-blue scale. Sidelines indicate statistical significance ($P < 0.05$). **e** Heat map for sound-sensitive genes after 24 h of sound emission, shown in the same way as in (**d**). **f** Correlations of $\log_2$ gene response values with different sound patterns. Genes that responded oppositely to the 440 Hz and 14 kHz sound in 24 h are indicated in red. **g** Biological activities significantly affected by sound emission, revealed by gene annotation analysis performed using Metascape[13]. Pathways with high statistical significance ($-\log_{10}P > 5$) are listed.

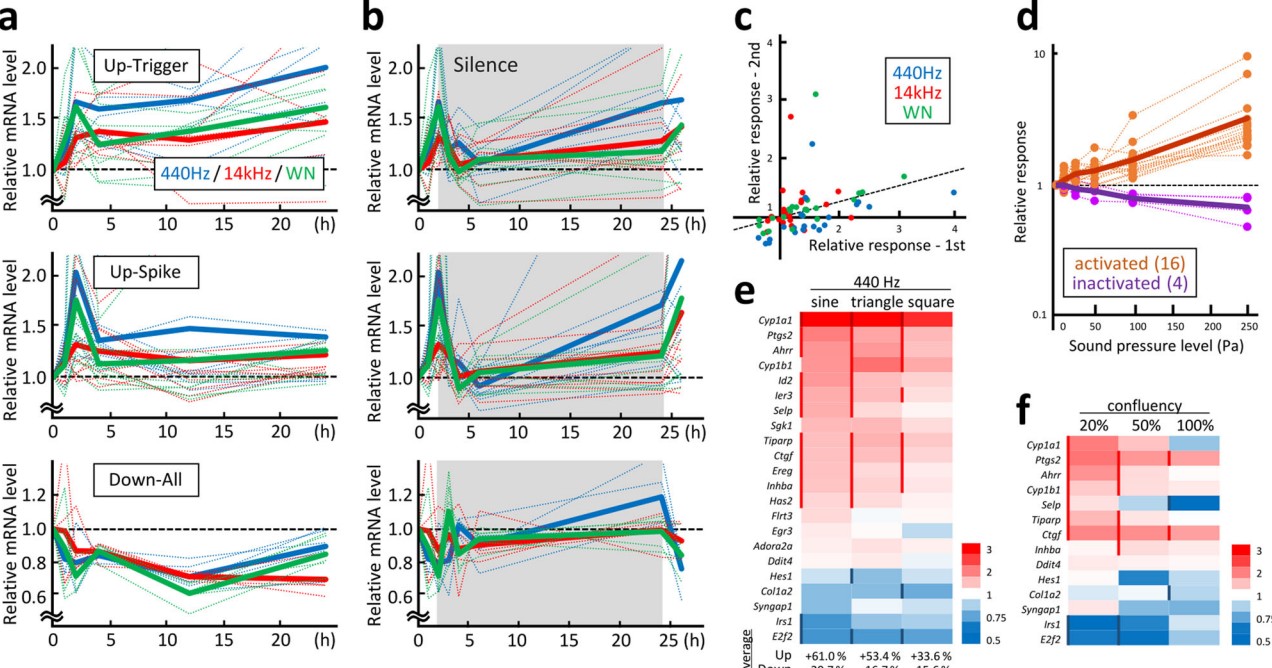

**Fig. 2 | Cellular and acoustic factors affecting gene responses. a** Time-course patterns of sound-sensitive gene responses analysed using quantitative PCR. Based on the response pattern to 440 Hz acoustic stimulation, genes were grouped into upregulated in the triggered pattern (8 genes), upregulated in the spiked pattern (9 genes), and downregulated (5 genes). Dotted lines indicate each gene, and thick lines indicate the averaged responses for continuous emission of 440 Hz (blue), 14 kHz (red) and white noise (green) sound. **b** Gene response patterns for silence and re-emission sound. After 2 h of emission, the sound was stopped for 24 h (highlighted in grey), and the same sound was re-emitted for 2 h. Grouping and data representation are the same as in (**a**). **c** Comparison of the responses for the 1st and 2nd sound emissions presented in (**b**). Relative responses for the 1st stimulation (0−2 h) and 2nd stimulation (24−26 h) are shown on the X- and Y-axes, respectively. The

linearly approximated line indicates a 2nd-to-1st response rate of 0.26 on average. **d** Gene responses at different sound pressure levels. The 440 Hz sound was emitted at 10, 25, 50, 100, and 250 Pa for 2 h. Dotted lines indicate activated (16 genes, orange) and inactivated (4 genes, purple) genes, and thick lines indicate the averaged responses. **e** Gene responses to different waveforms. Sine-, triangle-, and square-waved 440 Hz sound was emitted at 100 Pa for 2 h. The relative gene response is shown on a red-to-blue scale, and the sidelines indicate statistical significance ($P < 0.05$). The average values for the upregulated and downregulated genes are shown below. **f** Effect of cell confluence on sound-sensitive gene responses. The relative response is shown on a red-to-blue scale, and the sidelines indicate statistical significance ($P < 0.05$).

response genes (i.e., upregulated by one sound pattern and downregulated by the other). The heat map shows that the early-response genes responded similarly to the three sound patterns (Fig. 1d). However, several characteristically unique responses to the different sound patterns were observed after 24 h (Fig. 1e). In particular, a set of genes significantly upregulated by the 14 kHz sound was downregulated by the 440 Hz sound (indicated in red in Fig. 1f). This may be primarily due to the different fluid actions of the culture medium; the 440 Hz vibration caused convective mixing of the medium, while the 14 kHz high-frequency vibration induced much less flow, because the amplitude of particle displacement of sine wave is inversely proportional to the frequency. In our experimental system, a diaphragm covering 80% of the dish area was attached to the culture medium to efficiently generate acoustic waves. Taken together with the fluid action, 14 kHz stimulation generated a certain extent of hypoxia in the culture medium as many of the 14 kHz responsive genes were found to be involved in the cellular response to hypoxia (Supplementary Fig. 3a). Gene annotation analysis using Metascape[13] revealed various molecular, cellular, and body/tissue-level activities affected by acoustic stimulation (Fig. 1g). In addition to the known mechanosensitive activities, such as fluid shear stress response, cell migration, cell adhesion, and blood vessel development, various pathways and processes were identified including apoptosis, synaptic transmission, and rhythmic process that may be unique responses to acoustic stimulation.

### Cellular and acoustic factors involved in the response

The responses of the sound-sensitive genes at different time points and for variable sound patterns were further analysed. A set of early-response genes

was selected for further analyses using quantitative polymerase chain reaction (PCR). Gene responses were classified into two patterns when the cells were exposed to continuous 100 Pa acoustic stimulation: triggered-type genes maintained altered mRNA levels for hours, whereas spiked-type genes responded transiently and returned to their basal expression levels (Fig. 2a). The upregulated genes were separated into two types, whereas the downregulated genes were mostly found to be of the triggered type. A significant overlap was not observed between the functions of the triggered- and spiked-type genes (Supplementary Fig. 3b). Gene responses to growth factors are known to be transient, whereas those to hormones and apoptotic stimulation are slower and persistent[14–16]. This highlighted the dual effects of acoustic stimulation on both signal transduction events. Most gene activity returned to basal levels after 2 h when the cells were exposed to sound for only 2 h, followed by incubation in a silent environment (Fig. 2b). Meanwhile, most genes responded repeatedly when the sound was re-emitted after 24 h. The relative intensity of the primary and secondary responses was 1:0.26, demonstrating the repetitive nature of the sound-induced gene responses (Fig. 2c).

Frequency, intensity, and waveform are the three major factors that determine the properties of sound. Proportional responses were observed for sound-activated and inactivated genes after applying 440 Hz sound with 10–250 Pa intensity (Fig. 2d). Considering that the sound energy quantity proportionally correlates to the square of the sound pressure and amplitude, these gene responses were expected to occur owing to a mechanical sensing system, rather than the energy conversion process. The effect of different waveforms was also assessed using 440 Hz triangle and square waves, which are enriched in high-frequency components in addition to the basic 440 Hz

frequency (Supplementary Fig. 4). Similar gene response patterns were observed 2 h after sound emission when the sounds were emitted at an intensity of 100 Pa (Fig. 2e). The sine wave was slightly more efficient than the other waveforms, as revealed by the quantitative comparison of the averaged gene responses.

Cell density is an important cellular factor in this response. Indeed, several gene responses were abolished or even reversed at high or low cell densities when cells of different densities were prepared (Fig. 2f). Considering the cellular and acoustic factors affecting gene responses, we focused on prostaglandin-endoperoxide synthase 2 (*Ptgs2*, also known as cyclooxygenase-2 [*Cox-2*]) and connective tissue growth factor (*Ctgf*, also known as cellular communication network factor 2 [*Ccn2*]) as marker genes for sound-triggered gene responses, as they exhibited high and stable responses under different cellular and acoustic conditions.

We have previously shown that *Ptgs2* and *Ctgf* are suppressed by the indirect emission of acoustic waves by a loudspeaker system[12]. These genes are mechanosensitive and show opposite responses to opposing forces, such as stretching and compression[17,18]. In contrast to the simple direct sound emission system used in this study, the indirect loudspeaker system produced a complicated acoustic field, as the sound generated in the air was not only transmitted to the medium but also vibrated the dish or baseplate. To assess the sound transmission modes of the loudspeaker system, a sound-field simulation was performed with different combinations of water, dish, and baseplate. Comparison of the pressures observed at the detection point revealed that among the total sound reaching the cells, the sound transmitted from the air to the medium only accounted for ~40% (Supplementary Fig. 5). Therefore, the sound transmitted by a dish or baseplate is suggested to act as a suppressor of sound-responsive genes. To further test this postulation, the sound was emitted by a vibrational transducer without fixing the dish, allowing free vibrations of the dish. Under these conditions, the responses of *Ptgs2* were significantly reduced, supporting our interpretation that indirectly transmitted sound elicited opposing effects to directly generated sound on sound-responsive genes (Supplementary Fig. 6).

## Mechanisms involved in perception and transmission of acoustic stimulation

To understand the signal transduction mechanisms leading to gene responses, the cellular effects of acoustic stimulation were investigated. The vibrational transducer was mounted on a microscope stage with a heating system (Supplementary Fig. 7a). As the sound emission (especially at low frequencies) significantly disturbed the image quality of the laser-scanning microscope, several averaging processes were applied (Supplementary Fig. 7b). C2C12 cells expressing EGFP-fused Lifeact were subjected to 440 Hz sine-wave, 14 kHz sine-wave, or white noise emitted at 100 Pa. Time-lapse observations and tracing of the cell area revealed expansion of the cell edge in response to acoustic stimulation (Fig. 3a). In contrast to the filopodia-dominant cell edge fluctuation of the silent cells, the sound-irradiated cells exhibited a clear expansion with lamellipodial smooth edges. Quantification of the cell expansion and retraction revealed that both activities were activated by sound stimulation, suggesting the involvement of cytoskeletal remodelling factors related to cell migration and adhesion (Fig. 3b); this agrees with the results of the gene annotation analysis (Fig. 1g). All three sound patterns acted similarly to increase the cell adhesion area by 15–20% in 1 h (Fig. 3c).

Focal adhesion is a major cell adhesion structure connecting the cytoskeleton to the extracellular matrix and functions as a sensor of mechanical stimuli, such as physical forces and matrix rigidity[19]. Focal adhesion kinase (FAK) plays an essential role in mechanotransduction at focal adhesions by organising cytoskeletal dynamics and regulating adaptive gene expression and cell adhesion/migration activities[20–22]. Phosphorylation of Y397 of FAK stabilises focal adhesions and promotes the expansion of the lamellipodial cell edge[23]. Western blot analysis revealed a time-dependent increase in Y397 phosphorylation following acoustic stimulation (Fig. 3d). Cell expansion in response to acoustic

stimulation was abolished when the cellular response was observed in the presence of Y15—a specific inhibitor of FAK Y397 phosphorylation (Fig. 3e). Under this condition, the responses of *Ptgs2* and *Ctgf* were abrogated, demonstrating the upstream role of FAK phosphorylation in these gene responses (Fig. 3f).

The downstream events were also investigated. *Ptgs2* encodes a key enzyme that regulates the biogenesis of prostaglandin E2 ($PGE_2$) and is involved in the adaptive regulation of $PGE_2$ production. The amount of $PGE_2$ released in the medium was 1.2−1.8 times higher for 4 days when 440 Hz sound was continuously emitted than that in the silent environment (Fig. 3g). This effect was dependent on the acoustic response via FAK signalling, as Y15 abolished the effect of sound. Several sound-responsive genes (Fig. 1d, e) were identified as responsive to increased $PGE_2$ production. Gene responses similar to those in acoustic stimulation were observed without sound emission when comparable amounts of $PGE_2$ were added to the medium (Fig. 3h). L161.982—an inhibitor of the major $PGE_2$ receptor, EP4—, cancelled the effect of sound on these genes, demonstrating that *Ptgs2* activity is a trigger for a set of sound-responsive genes. Furthermore, L161.982 treatment did not affect cell morphological responses to sound, whereas adding $PGE_2$ alone to the medium did not induce cell adhesion (Fig. 3e).

Taken together, the acoustic signal transduction pathway is initiated at focal adhesions, induces *Ptgs2* response via FAK phosphorylation, and increases $PGE_2$ activity against the EP4 receptor, activating a gene set (Fig. 4a). Consequently, the cells reinforced focal adhesions and expanded their adhesion areas.

The cell-type specificity associated with the activation of this pathway by acoustic stimulation was examined using several cell lines, including epithelial, stromal, fibroblast, and neuroblast cells. The effect of acoustic stimulation on FAK Y397 phosphorylation, *Ptgs2* expression, and $PGE_2$ biogenesis was evaluated (Fig. 4b and Supplementary Fig. 8), revealing a markedly higher response in 3T3-L1 preadipocytes.

## Suppression of adipocyte differentiation by acoustic stimulation

Adipocyte differentiation is affected by $PGE_2$ via EP4 receptor signalling[24,25]. *Ptgs2* expression is suppressed in developed white adipose tissues, which is directly related to reduced $PGE_2$ production and increased fat mass in the tissue[26]. Therefore, adipocyte differentiation is directly affected by $PGE_2$ production under the control of *Ptgs2*, suggesting the potential for regulating adipocyte activity through acoustic stimulation.

Adipocyte differentiation cultures using 3T3-L1 cells were subjected to acoustic stimulation. When the cell density reached ~70%, the medium was replaced with methylisobutylxanthine, dexamethasone, insulin (MDI)-containing differentiation induction medium and incubated for 3 days, followed by incubation in differentiation-enhancing medium containing insulin alone for four additional days; 440 Hz sine-wave sound was emitted at 100 Pa during the 3-day culture in MDI differentiation induction medium according to the following schedule: continuous emission during days (D) 0–3 (72 h), D0-1 (24 h), D1-2 (24 h), and D2-3 (24 h) or periodic emission for 2 h every day (2 h × 3; Fig. 5a). The effect of acoustic stimulation on adipocyte differentiation was evaluated by determining the expression levels of the marker genes, CCAAT/enhancer-binding protein α (*Cebpa*) and peroxisome proliferator-activated receptor γ (*Pparg*). Quantitative PCR analyses revealed considerable suppression of both genes on D3 in the D0–3 sample, exhibiting relative expression levels of 0.18 and 0.26 for *Cebpa* and *Pparg* (Fig. 5b). The 2 h × 3 sample exhibited approximately 30% suppression of both genes. The D0-1, D1-2, and D2-3 samples did not show statistically significant suppression of either gene, suggesting the need for persistent acoustic stimulation to achieve an efficient cell response.

The cells were completely differentiated under the above-mentioned culture conditions for 7 days, with or without acoustic stimulation. After D4, acoustic stimulation could not be applied owing to attenuated cell

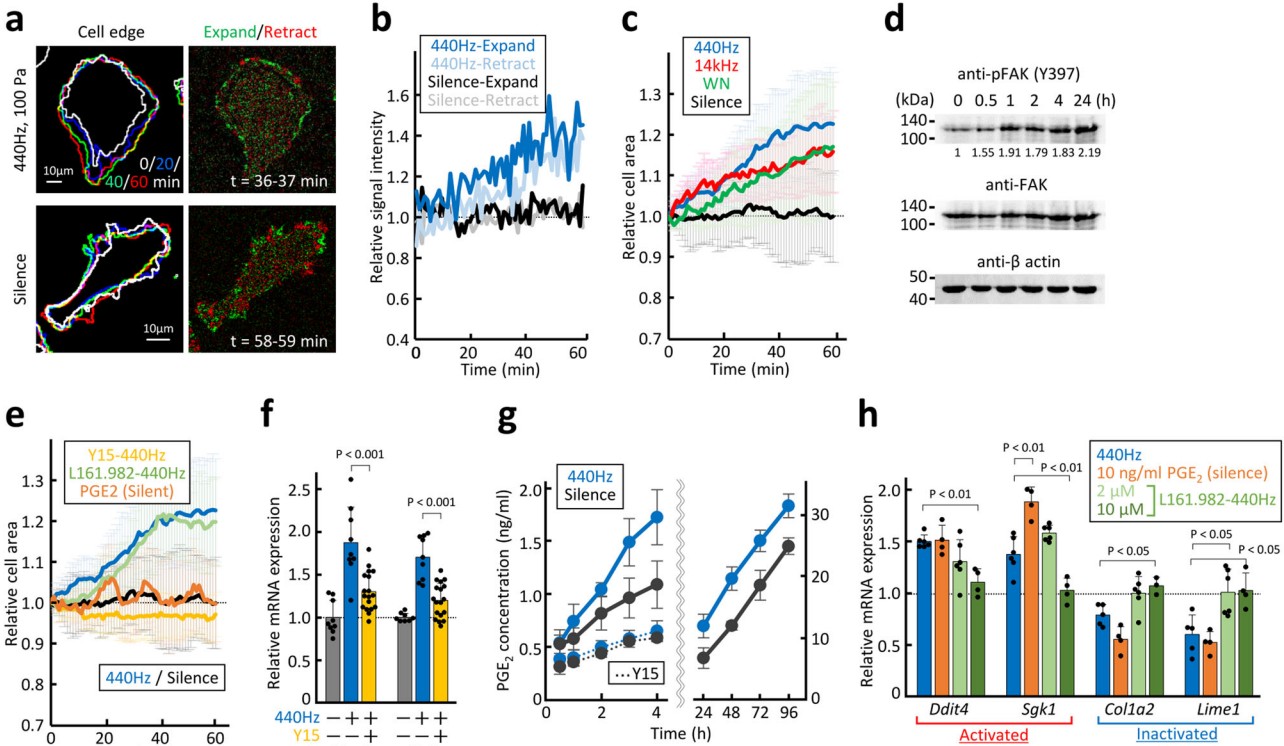

**Fig. 3 | Pathways involved in the perception and transmission of acoustic stimulation. a** Live observation of C2C12 cells expressing EGFP-Lifeact under acoustic stimulation with 440 Hz sine-wave sound at 100 Pa. Image processing procedures are described in Supplementary Fig. 7. Cell edges at time 0, 20, 40, and 60 min are overlaid in white, blue, green, and red, respectively. Expanded and retracted areas in 1 min are shown in green and red, respectively. Results are shown in the movies in Supplementary Movies 1–4. **b** Time-course changes in relative intensity of signals of expanding/retracting areas under 440 Hz stimulation (blue/light blue) or silence (black/grey). **c** Time-course changes in relative cell area at 440 Hz (blue), 14 kHz (red), and white noise (green) stimulation at 100 Pa or under silence (black). $n ≥ 5$ for each condition, and vertical bars indicate $± SD$. **d** Western blot analysis of C2C12 whole-cell lysate taken at the indicated time point after starting 440 Hz continuous acoustic stimulation at 100 Pa. After blotting with an anti-phospho Y397 FAK antibody, the membrane was stripped and reblotted with an anti-FAK antibody. Anti-β actin blotting was performed as a loading control. Normalised signal intensities of phosphorylated FAK relative to those of total FAK are indicated. Unedited original blot images are shown in Supplementary Fig. 11. **e** Time-course

changes in the relative cell area under 440 Hz (blue) sound, silence (black), 440 Hz sound with 2 μM Y15 (yellow) and 10 μM L161.982 (light green), and silence with 10 ng/ml $PGE_2$ (orange). $n ≥ 5$ for each condition, and vertical bars indicate $± SD$. **f** Expression levels of *Ptgs2* and *Ctgf* after 2 h emission of 440 Hz sound at 100 Pa in the presence (yellow) or absence (blue) of 2 μM Y15 relative to those of the silent condition (grey). Bars represent $+SD$ from 3 biological replicates of ≥3 independent experiments, and statistical significance was evaluated using one-way ANOVA followed by Tukey's HSD test. **g** Amount of $PGE_2$ released into the culture medium was quantified using a chemiluminescence assay under 440 Hz continuous stimulation at 100 Pa (blue) or silence (grey) at different time points. Dashed line indicates results in the presence of 2 μM Y15, added 1 h before starting the sound stimulation. Bars represent $± SD$ from $n ≥ 3$. **h** Responses of four sound-sensitive genes in the presence of 10 ng/ml $PGE_2$ (orange) or 2/10 μM L161.982 with 440 Hz stimulation (green). Bars represent $+SD$ from three biological replicates of two independent experiments, and statistical significance compared with 440 Hz sound alone (blue) was evaluated using one-way ANOVA followed by Tukey's HSD test.

adhesion along with adipocyte differentiation, often resulting in the detachment of cells from the culture dish. Lipid accumulation was monitored by staining the cells with the nonpolar fluorescent dye, BODIPY 493/503, and classified into undifferentiated (no lipid droplet), early (~several micrometre lipid droplets), and late (many lipid droplets >10 μm in size) stages of adipocyte differentiation (Fig. 5c). When D0–3 acoustically stimulated cells were cultured until D7, the ratio of cells that remained undifferentiated increased significantly from 0.23 to 0.39 compared with that under the silent condition (Fig. 5d). Periodic 2 h × 3 stimulation also efficiently increased the ratio of the undifferentiated cell population to 0.43, whereas none of the one-day stimulations significantly altered the differentiation states. The effects of D0–3 and 2 h × 3 were comparable to those observed after the addition of 1000 ng/ml $PGE_2$ to the MDI medium. The amount of lipids accumulated in the cells was quantified by measuring the fluorescence intensity of the BODIPY 493/503 dye extracted with 2-propanol after staining. The relative amount of the signal revealed that D0–3 and 2 h × 3 acoustic stimulations suppressed lipid accumulation by 13−15% (Fig. 5e), equivalent to that observed after adding 1000 ng/ml $PGE_2$ and consistent with the microscopy results (Fig. 5d). The increase in the $PGE_2$

concentration due to acoustic stimulation remained at <100 ng/ml even after 3 days (Fig. 3g). Thus, the high efficiency of acoustic stimulation in suppressing adipocyte differentiation is indicative of the involvement of other pathway(s) in acoustic signal transduction, which are yet to be identified.

## Discussion

In this report, we revealed the properties, mechanisms of action, and effects of cell-level responses to the audible range of acoustic stimulation. All three sound factors (frequency, intensity, and waveform) affected gene response patterns, albeit in slightly different manners. Comparisons of 440 Hz and 14 kHz sound effects indicated that the early gene responses were similar, whereas frequency-specific responses were more prominent later (Fig. 1c–f). The effects on cell morphology within an hour were also similar among different frequencies (Fig. 3c), suggesting a cytoskeleton-linked shared sound response mechanism for the early-response and unknown mechanisms that trigger frequency-specific cell responses in longer timeframes. Early-response genes were more enriched in those involved in cell migration and adhesion (Fig. 1g), further supporting this idea. Evaluation of the gene responses at shorter time

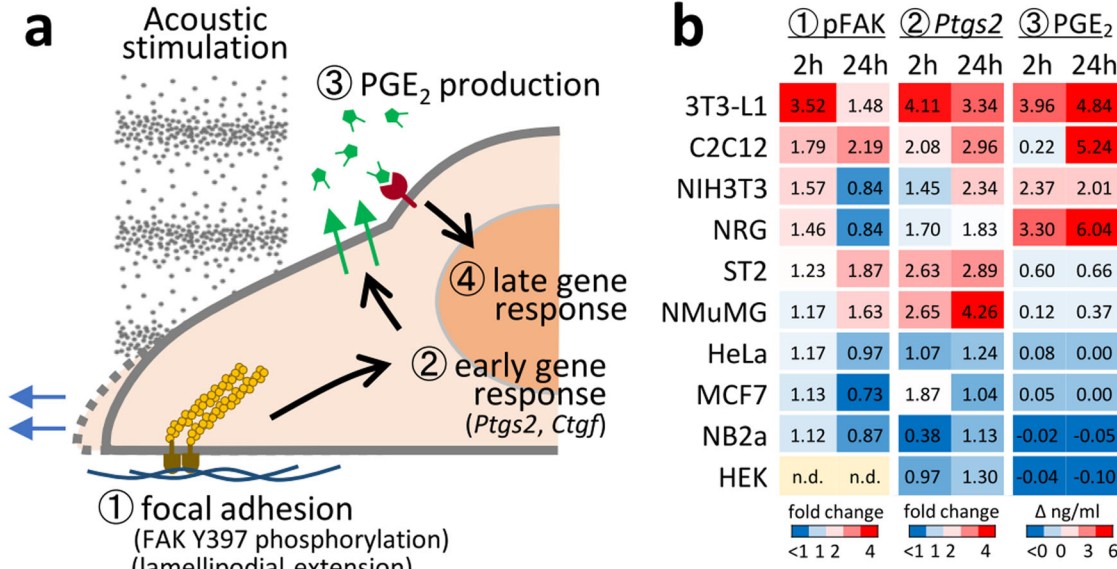

**Fig. 4 | Pathway model and cell-type specificity of acoustic signal transduction. a** Model of the acoustic signal transduction pathway revealed in this study. **b** Cell-type specific response to acoustic stimulation. In addition to C2C12 myoblast derived from muscle tissue, 3T3-L1 (fibroblast, embryo), HEK (epithelial, kidney), HeLa (epithelial, cervix), MCF7 (epithelial, breast), NB2a (neuroblast, neuroblastoma), NIH3T3 (fibroblast, embryo), NMuMG (epithelial, breast), NRG (fibroblast, bone marrow), ST2 (fibroblast, bone marrow) cells were subjected to 440 Hz acoustic stimulation at 100 Pa intensity for 2 and 24 h. Phosphorylation states of FAK Y397 were analysed by quantifying pFAK/FAK signal intensities of Western blotting using anti-FAK and anti-pFAK antibodies, and the effect of acoustic stimulation was evaluated over silent condition. Blue-to-red indicates fold change from <1 to 4. Anti-pFAK signal was not detected for the HEK cell sample (n.d.). Expression of *Ptgs2* was analysed by quantitative PCR and evaluated over silent condition. Blue-to-red indicates fold change from <1 to 4. Concentration of PGE₂ in the culture medium was measured by PGE₂ CLIA kit, and the increase by the acoustic stimulation (Δng/ml) was shown in blue-to-red indicating <0 to 6 ng/ml increase of PGE₂. Detailed data were presented in Supplementary Fig. 8.

points revealed that 2 h was a reasonable timeframe to detect the expression of early-response genes, as two marker genes *Ptgs2* and *Ctgf* showed statistically significant responses in 1–2 h after the acoustic stimulation (Supplementary Fig. 9).

The generation of acoustic waves in water inevitably accompanies fluid movement. Although it is impossible to experimentally assess the sole effect of the compressional wave free of shear stress caused by the flow, comparison of the cell responses for 440 Hz and 14 kHz sound waves strongly suggests that many of the cellular responses observed in this study were induced by the compressional waves. When comparing acoustic waves with the same intensity, the amplitude of particle displacement of the sine wave is inversely proportional to the sound frequency. Thus, under the same 100 Pa output, a 440 Hz sound wave will have a 32-fold larger particle displacement amplitude than that of a 14 kHz sound wave and generate much larger shear stress for the cells. Therefore, cell responses caused by the shared stress are expected to be much higher at 440 Hz than those at 14 kHz. The correlation analyses of gene responses in response to 440 Hz and 14 kHz sound waves revealed dispersed distribution (Fig. 1f), with approximately half of genes (45.5%) showing stronger response to 14 kHz stimulation than to 440 Hz stimulation (Supplementary Fig. 10a). Histogram showing the differences in the absolute response at 440 Hz from that at 14 kHz revealed normal distribution with the peak value of 0.06 (Supplementary Fig. 10b). Therefore both 440 Hz and 14 kHz stimulation induced similar levels of unique gene responses, implying that compressional wave is the major cause of these gene responses. This interpretation is also supported by an annotation analysis. Among 254 genes annotated as "response to fluid shear stress (GO:0034405)" in the gene ontology database[27], only 3 genes overlap with sound-sensitive genes identified in this study (Fig. 1d, e). This annotation category was not found in the top 40 annotation clusters enriched in the sound-sensitive genes, suggesting a minimal influence of the fluid shear force in our acoustic experimental condition. Notably, larger effects of the fluid movement

may be included in the acoustic stimulation at lower frequencies in general. Further studies comparing different frequencies and output intensities will be useful to precisely estimate the effect of the fluid movements and elucidate the differences in results from those of compressional wave.

The effects of the different waveforms may be linked to different frequencies. Triangular and square waves contained more high-frequency overtone series than the pure sine wave (Supplementary Fig. 4); therefore, cell responses to triangular or square waves may include the effect of high-frequency waves. Comparison of the gene responses to 440 Hz and 14 kHz sound waves revealed higher correlation for square and triangular waves than sine wave, with the correlation coefficients of 0.76, 0.74, and 0.68 for square, triangular, and sine waves, respectively against 14 kHz, supporting this interpretation.

Considering the nature of acoustic waves and certain overlaps between sound-sensitive and mechanosensitive genes, the sound response mechanism is assumed to partially involve mechanotransduction. Two sound response marker genes identified in this study, *Ptgs2* and *Ctgf*, are mechanosensitive and are involved in responses to forces derived from cell-matrix interactions, such as stretch, compression, fluid shear stress, and matrix rigidity[28–30]. FAK is also involved in mechanosensing and regulates cell migration via Y397 phosphorylation, as observed in migrating fibroblasts[31]. In general FAK-dependent mechanical sensing is dependent on cell confluence, as FAK is mainly involved in cell-matrix communication in a sparse condition, which is replaced to cell-cell communication at high confluence[32]. FAK-dependent gene responses such as *Ptgs2* and *Ctgf* under the condition of low as well as high confluence (Fig. 2f) imply a novel form of FAK-dependent signal transduction triggered by acoustic stimulation. Taken together with the gene and cell morphological responses (Figs. 1 and 3), the effect of acoustic stimulation may be similar to that of a stiff matrix. Our observation that acoustics negatively impact adipocyte differentiation (Fig. 5) supports this idea, as a stiff matrix also suppresses adipocyte differentiation[33].

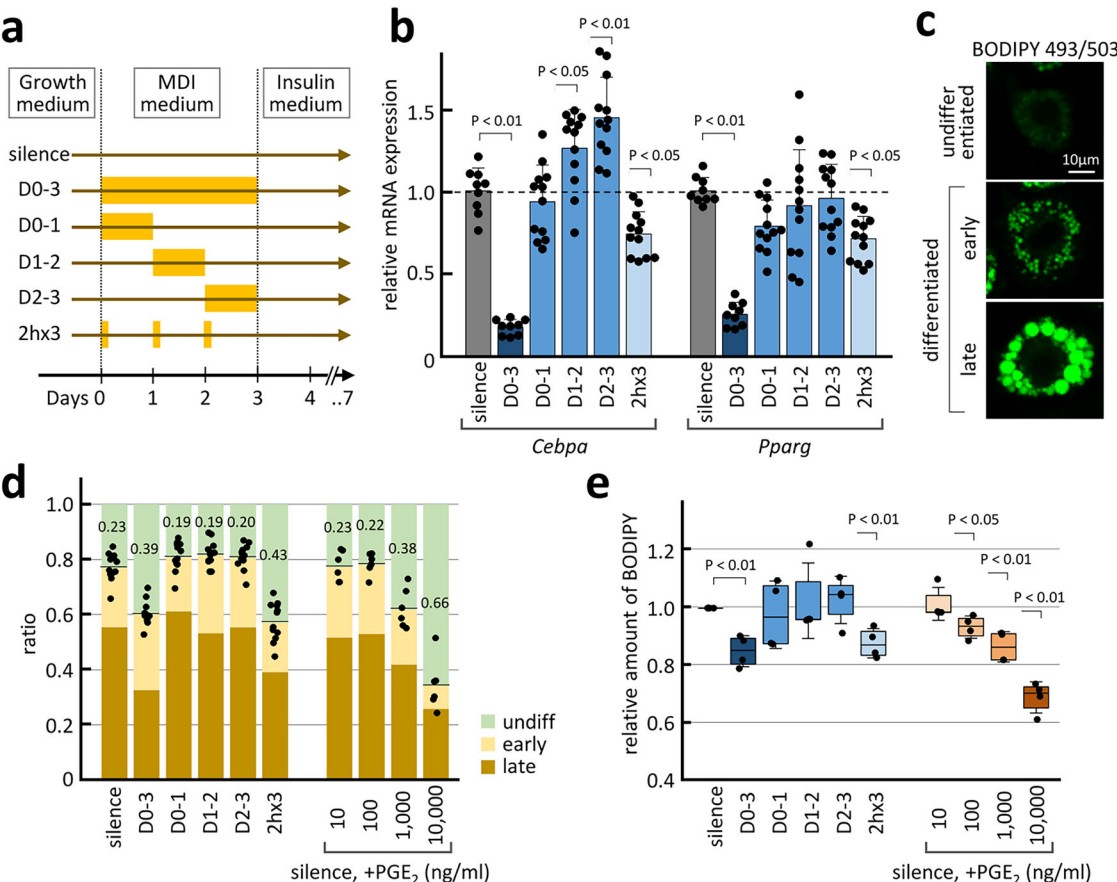

**Fig. 5 | Effect of acoustic stimulation on adipocyte differentiation. a** Time course of the acoustic stimulation applied in this study. Stimulation periods are indicated in yellow. **b** Relative mRNA expression levels of adipocyte differentiation marker genes, *Cebpa* and *Pparg*, on differentiation day 3 of 3T3-L1 cells as revealed using quantitative PCR analyses. Bars represent +SD from 3 biological replicates of ≥3 independent experiments, and statistical significance compared with silence (grey) was evaluated using one-way ANOVA followed by Tukey's HSD test. **c** Lipid staining images of 3T3-L1 cells labelled with BODIPY 493/503. Images of undifferentiated, early differentiated, and late differentiated cells are shown. **d** Ratio of undifferentiated (green), early differentiated, and late differentiated (light and dark

brown) 3T3-L1 cells on day 7. Acoustic stimulation conditions or $PGE_2$ concentrations supplemented in the MDI medium are indicated. Data from three biological replicates of three (acoustic stimulation) and two ($PGE_2$ stimulation) independent experiments are presented. Dots represent each data point for the border of undifferentiated and differentiated cells. The ratio of undifferentiated cells is indicated. **e** The relative amount of lipid accumulation in day 7 cells was quantified by measuring the fluorescence intensity of BODIPY 493/503 extracted in 2-propanol after cell staining. Box indicates an average and 25–75 percentiles; bars represent ±SD from four independent experiments; statistical significance compared with the silent sample was evaluated using Welch's *t*-test.

Low-intensity pulsed ultrasound (LIPUS) at several MHz frequencies also induces *Ptgs2* and *Ctgf* expression in bone cells, such as osteoblasts and chondrocytes[34,35]. Several studies have reported the effects of LIPUS in suppressing adipocyte differentiation. In response to LIPUS transmitted at different MHz frequencies, three cellular pathways are reportedly activated: ERK signalling through Rho-associated kinase or insulin receptor signalling[36,37], YAP nuclear translocation[37], and histone deacetylase 1[38], all resulting in the suppression of adipocyte differentiation. The focal adhesion-mediated pathway revealed in this study is different from the pathway activated in response to LIPUS, which is expected, considering an approximately $10^4$-fold difference in the frequency between LIPUS and audible sound sources used in this study. Several genes showed unique and characteristic responses to different sound stimulations (Fig. 1, Supplementary Fig. 3), suggesting a unique feature of acoustic stimulation that differs from other mechanical stimuli. Considering the infinite sound pattern with both temporal and compositional variations, acoustic stimulation may induce diverse cellular responses and, therefore, is an intriguing tool for cell manipulation such as living tissue engineering, regenerative medication, artificial tissue culture and related biotechnology industry.

Differences in sensitivity to acoustic stimulation in different cell types likely reflect structural and functional heterogeneity of the focal

adhesion. The size, number, and molecular components of the focal adhesions vary significantly and are closely related to the cell motility and adhesion properties[39]. Adhesive stromal and its derivative cells, including fibroblasts, myoblasts, osteoblasts, and adipocytes, were highly sensitive to acoustic stimulation (Fig. 4b), possibly due to their highly adhesive and motile nature that is essential to develop active focal adhesions. In contrast, epithelial and neuroblastoma cells, which are less mobile or less adhesive, were relatively insensitive. Although this focal adhesion-dependent pathway preferentially acts in stromal cell lines, other sound perception pathways may be active in other cell types. Further studies comparing the global gene expression patterns of different cell lines in response to different sound patterns unveil mechanisms underlying cell responses to sound.

Sound exists universally in the material world; thus, life forms are exposed to this physical energy from their inception. It is, therefore, not surprising that living organisms develop systems to utilise sound as an environmental stimulus and optimise activities at the cellular level. Our findings reveal the nature of cells that sense acoustic information to orchestrate inner activities and mark the beginning of research focusing on the fundamental relationship between life and sound.

## Methods

### Direct sound emission system

The detailed setup is shown in Supplementary Fig. 2. In short, a vibrational transducer (Mini-shaker Type 4810, Brüel & Kjær, Nærum, Denmark) attached to a laboratory scissor jack was inversely fixed to a small metal shelf. A diaphragm custom-made with PEEK (Nippon Chemical Screw, Tokyo, Japan) was selected by comparing the properties of a plate made of several different materials. The sound signal was output using a conventional MP3 player, adjusted by an amplifier (AP15d, Fostex, Tokyo, Japan), and input into the transducer. The position of the head of the diaphragm was adjusted for attachment to the culture medium to generate the acoustic waves directly in the medium. This system was primarily used in this study.

### Indirect sound emission system

A full-range active loudspeaker (6301NB, Fostex, Tokyo, Japan) was set under a small metal shelf. The sound signal was output using a conventional MP3 player, adjusted by an amplifier (AP15d, Fostex, Tokyo, Japan), and input into the loudspeaker.

### Sound sources

Single-frequency and noise sound signals were generated using the Tone Generator software (NCH Software, Canberra, Australia), with a sampling rate of 44,100 Hz. A total length of 60 s was repeatedly emitted. All sound data used in this study are available in Supplementary Audio 1–5.

### Sound intensity measurement in water

Sound was acquired in water using a hydrophone (AQH-100, Aqua-Sound, Kobe, Japan) and processed using an Aquafeeler monitoring system (AquaSound, Kobe, Japan) and a digital converter (SE-U33GX, Onkyo, Osaka, Japan). The signal was recorded and analysed using the SP4Win custom software (NTT Advanced Technology Corporation, Tokyo, Japan) to obtain the root mean square (RMS) value. Considering the sensitivity of the hydrophone to be -207.3 dB re. 1 V/µPa, it detected 15,311 Pa/V. The output with 1 V of a 1 kHz sine-wave standard signal from the Aquafeeler was used for calibration, according to which 1 V = 19929.2 RMS. The sound pressure obtained was 0.768 Pa/RMS using this system. The pressure level obtained under the silent condition was subtracted as the background noise.

### Cell culture, drugs, PGE$_2$ quantification, and western blotting

C2C12 (RCB0987), NIH3T3 (RCB2767), MC3T3-E1 (RCB1126), NRG (RCB1921), ST2 (RCB0224), and NB2a (RCB2639) were obtained from Riken Bioresource Research Center (Ibaraki, Japan), and 3T3-L1 (JCRB9014) was obtained from Japanese Collection of Research Bioresources (Osaka, Japan). The cells were cultured in Dulbecco's modified Eagle's medium (DMEM; D6046 and D6429, Sigma-Aldrich, St. Louis, MO, USA) supplemented with 10% foetal bovine serum (BCBZ5443, Sigma-Aldrich, St. Louis, MO, USA), penicillin-streptomycin (26253-84, Nacalai Tesque, Kyoto, Japan), and amphotericin B (SV30078.01, Cytiva HyClone, MA, USA) in a water-jacketed incubator supplied with 5% CO$_2$. The cells were carefully maintained at a maximum of 60% confluence to avoid the effect of contact inhibition and spontaneous differentiation. Thereafter, 2 µM Y15 (S5321, Selleck, Houston, TX, USA), 2 or 10 µM L161.982 (SML0690, Sigma-Aldrich, St. Louis, MO, USA), and 10−10,000 ng/ml PGE$_2$ (29334-21, Nacalai Tesque, Kyoto, Japan) were added when required. The amount of PGE$_2$ in the culture medium was quantified using a PGE$_2$ CLIA kit (ADI-910-001, Enzo Life Sciences, Farmingdale, NY, USA) according to the manufacturer's instructions. Western blotting was performed using anti-phospho Y397 FAK (ab81298, Abcam, Cambridge, UK), anti-FAK (13009, Cell Signaling Technology, Danvers, MA, USA), and anti-β-actin (A5441, Sigma-Aldrich, St. Louis, MO, USA) antibodies. Horseradish peroxidase-conjugated anti-rabbit IgG (A27036, Invitrogen/Thermo Fisher Scientific, Waltham, MA, USA) and anti-mouse IgG (Cytiva, MA, USA) were used as secondary antibodies and

detected using the Chemi-Lumi One L reagent (07880, Nacalai Tesque, Kyoto, Japan) and LAS-3000 mini imaging system (FujiFilm, Tokyo, Japan).

### RNA-sequencing analysis

C2C12 cells in a 30 mm dish that were ~40% confluent were treated with a 440 Hz sine-wave sound, 14 kHz sine-wave sound, and white noise at 100 Pa. After continuous sound emission for 2 and 24 h, total RNA was extracted using an RNeasy kit (74104, Qiagen, Hilden, Germany). The RNA library was prepared using the NEB Next Ultra II Directional RNA library prep kit for Illumina (E7760, New England BioLabs, Ipswich, MA, USA). RNA-sequencing was performed using NextSeq500 (Illumina, San Diego, CA, USA), with a single-end read and high output for 75 cycles. Trimming of noise signals (poly G), low-quality reads (~30 bases), and an additional base at the 76th position was performed using Cutadapt[40] and Trimmomatic[41]. Mapping and identification of differentially expressed genes were performed using the CLC Genomics Workbench 21 software (Qiagen, Hilden, Germany). In short, reads were mapped on the mouse genome sequence (GRCm38.91) and normalised by the trimmed mean of M values method. Samples from three independent experiments were analysed. Genes with >1.2 fold change (maximum false discovery rate [FDR]-$P < 0.05$) in 2 h sound-stimulated samples were selected as early sound response genes, and those with >1.5 fold changes (FDR-$P < 0.05$) after 24 h were selected as late sound response genes. Annotation analysis was performed using the Metascape gene annotation tool[13] with the list of sound response genes.

### Reverse transcription-coupled quantitative PCR (RT-qPCR) analysis

Total RNA was extracted using the RNeasy kit (74104, Qiagen, Hilden, Germany) and used as a template for RT-qPCR using a One Step SYBR PrimeScript Plus RT-PCR kit (RR096B, Takara Bio, Shiga, Japan) and an RT-qPCR system (LightCycler 480 and LightCycler 96, Roche, Basel, Switzerland). The primer sequences used to detect mouse mRNAs are listed in Table 1. Melting curve analysis was performed to assess the quality of the amplicons, and only single-peak runs were selected. The amounts of loaded templates under each condition were calibrated using the *Actb* mRNA level, and values relative to the mRNA levels from the silent condition were obtained.

### Microscopic observation and processing

The direct sound emission system was mounted on the stage of a confocal laser-scanning microscope (FV-1200, Olympus, Tokyo, Japan) equipped with a heating stage. The diaphragm was inserted into the heating chamber through a hole in the lid and covered with aluminium foil to stabilise the temperature (Supplementary Fig. 7). To cancel the effect of sound vibrations disturbing the image acquisition, three line scans and four z-stack projections were averaged. Images were acquired every minute for 1 h. To detect the cell edge, the images were binarised and traced. To obtain the expanded and retracted areas, each consecutive image was subtracted and presented as green and red, respectively. The ImageJ software[42] was used to adjust, binarise, subtract, and quantify the signals.

### Adipocyte differentiation and evaluation

3T3-L1 cells were cultured to ~70% confluence and incubated with MDI differentiation induction medium containing 0.5 mM methylisobutylxanthine (ab120840, Abcam, Cambridge, UK), 1 µM dexamethasone (ab120743, Abcam, Cambridge, UK), and 10 µg/ml insulin (ab123768, Abcam, Cambridge, UK) for 3 days. The medium was then replaced with a differentiation-enhancing medium containing 10 µg/ml insulin alone for four additional days. For lipid staining, the cells were incubated with 0.5 mM BODIPY 493/503 (D3922, Invitrogen/Thermo Fisher Scientific, Waltham, MA, USA) in serum-free DMEM for 5 min, washed twice with phosphate-buffered saline, and observed under a confocal laser-scanning microscope (FV-1200, Olympus, Tokyo, Japan). The BODIPY-stained cells were

**Table 1 | Sequences for primers used in qPCR analysis**

|  | Forward (5′ to 3′) | Reverse (5′ to 3′) |
|---|---|---|
| *Mus musculus* |  |  |
| *Ptgs2* | TGCACTACATCCTGACCCACT | TCTGGATGTCAGCACATATTTCA |
| *Ctgf* | CACACCGCACAGAACCAC | TTCATGATCTCGCCATCG |
| *Cyp1a1* | ATTCTGGCACAGAGGTGCTC | AAACCATTTGGGAAGGCTGT |
| *Ahrr* | AGCATAGATTTTCTTTCCTTCACC | CTTTCACTTGGCCAATGGTT |
| *Cyp1b1* | TTCGCCTCTTTCCGTGTG | GCTCAGAGTAGTGACCGAACG |
| *Id2* | ACTCGCATCCCACTATCGTC | TGGACGCCTGGTTCTGTC |
| *Ier3* | CTATGCGCTGGATCTTAAAGC | ACCCATCGCGTTTAGAAGG |
| *Selp* | TTGCTGAAAACGGGGAGAGTC | TCCCAAGAGAAAGACCTTCG |
| *Sgk1* | TCTCTTCCTTCCAACGTGGT | GTCTCAGGCGGCACTCTC |
| *Tiparp* | CAGAACAGGGGGTTCCAAT | CCACTGTCCCACTGATGGTT |
| *Ereg* | ATCAGCACAACCGTGATCC | TCCATCTGAACTAAGGCGGTA |
| *Inhba* | CAAAAAGGACATTCAAAACATGA | TCCATTTTCTCTGGGACCTG |
| *Has2* | CACACACACACCTTAGCTCCTC | ACCCCCATTGAATGTCTTTG |
| *Flrt3* | CTGGCTTATATGAGATGCTTGAAC | CCATTATGTAATATCTGGAAACTGTGA |
| *Egr3* | ATGTGACCGGAGGAGATGG | CATCCGAGGCATTTCTGTAAG |
| *Adora2a* | CAGGGCTATCTCCCGCTAAT | GCACCCTGCCTTTCATAGTT |
| *Ddit4* | CCTAGCCTCTGGGATCGTTT | CGGAGTTCGAGACGAGGAC |
| *Hes1* | CCTTTCTCATCCCCAACG | TCCCACTGTTGCTGGTGTAG |
| *Col1a2* | AGCCCTGGTTCTCGAGGT | CAGGAGGACCCATTACACCA |
| *Syngap1* | GATCCTGATGCAGTACCAAGC | TTCTCCACCTGCTGCTGTC |
| *Irs1* | GCGCAGTTACCTCGTCCTT | AAATTCTGACTCCAAAATTCACG |
| *E2f2* | ATGTCAGGCTGAGTCCCTTC | TTACAAACTCCCCCAAACTCC |
| *Lime1* | GCAGAGACTGTGGGAGCAA | TGACGCCTCACCTCACTTC |
| *Cebpa* | GCAAAGCCAAGAAGTCGGTGGA | CCTTCTGTTGCGTCTCCACGTT |
| *Pparg* | GTGAAGCCCATCGAGGACAT | ATCTTCTGGAGCACCTTGGC |
| *Actb* | GCCAACCGTGAAAAGATGAC | GAGGCATACAGGGACAGCAC |
| *Homo sapiens* |  |  |
| *Ptgs2* | TGGGAAGCCTTCTCTAACCTC | TCAGGAAGCTGCTTTTTACCTT |
| *Actb* | TCCAAATATGAGATGCGTTGTT | TGCTATCACCTCCCCTGTGT |

Sets of forward and reverse primers designed to detect each mRNA from mouse- and human-derived cells.

incubated with 100% 2-propanol for 5 min to extract the dye, and the fluorescence intensity was measured using a plate reader (TriStar 3, Berthold, Baden-Württemberg, Germany).

## Simulation analysis of acoustic transmission in an elastic object via physical contact

An elastic object of $0.001 \, m^3$ volume with a $1 \times 10^5 \, N/m^2$ shear elastic constant—stiffness typical for soft biological tissues used in simulation analyses[43]—was subjected to physical contact with an impact ball (YI-01, Rion, Tokyo, Japan) that provides maximum impact force corresponding to that of trotting male humans[44]. The analysis was performed based on the vibroacoustic finite-difference time-domain method[45].

## Simulation analysis of acoustic propagation using an indirect sound emission system

The sound source and detection points were set at an acoustic field of $252,000 \, mm^3$. The detection point was located at the centre of the water, and the sound source was located 50 mm from the detection point. The culture dish was made of acrylic plastic, and the Young's modulus, Poisson's ratio, and density were set to 3.2 GPa, 0.35, and $1190 \, kg/m^3$, respectively. The analysis was performed based on the finite element method using

commercially available software (COMSOL Multiphysics 6.1, Comsol, Burlington, MA, USA).

## Statistics and reproducibility

The normality of the data distribution was evaluated using the Shapiro–Wilk test. One-way ANOVA followed by Tukey's honestly significant difference (HSD) test was applied for normally distributed samples, and non-parametric Welch's *t*-test was applied for non-normally distributed samples to evaluate statistical significance. Sample sizes and number of replicates were described in each figure legend.

## Reporting summary

Further information on research design is available in the Nature Portfolio Reporting Summary linked to this article.

## Data availability

All the data supporting the findings of this study are available in this article and the Supplementary Information file. All the source data can be obtained in the Supplementary Data file. The RNA-sequencing data are publicly available in the Gene Expression Omnibus (GEO) database under the accession code GSE247726.

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

## Acknowledgements

We thank T. Kondo and Y. Sando for performing the RNA-sequencing analyses. This work was supported by SPIRITS (to M.K.) and FY2021 Kusunoki125 (to M.K.) from Kyoto University, JSPS KAKEN-HI Grants JP20K21389 (to M.K.) and JP22H05171 (to S.H.Y.), JST PRESTO Grant JPMJPR24O6 (to M.K.), The Murata Science Foundation H31-028 (to M.K.), and The Mitsubishi Foundation 201911006 (to M.K.).

## Author contributions

M.K. designed and performed the experiments, analysed the data, and wrote a draft of the manuscript. M.O. contributed to establishing the sound emission systems, recording and analysing the sound data, and interpreting the results. M.T. contributed to sound simulation analyses and interpreted the results. S.H.Y. supervised the study. All authors contributed to writing the manuscript.

## Competing interests

The authors declare no competing interests.
