## [Transparent Peer Review file · Communications Biology]

Acoustic modulation of mechanosensitive genes and adipocyte differentiation

Corresponding Author: Dr Masahiro Kumeta

This manuscript has been previously reviewed at another journal. This document only contains information relating to versions considered at Communications Biology.

Version 0:

Reviewer comments:

Reviewer #1

(Remarks to the Author)

In this manuscript, Kumeta et al. investigated cellular responses to audible sound using RNAseq and qPCR methods. They further explored the underlying sound-sensitive mechanisms and demonstrated the suppressive effects of acoustic stimulation on adipose differentiation. This study is an extension of their earlier paper published in 2018, where they examined cellular responses to "indirect audio sound" generated from a speaker. In this study, they modified their sound generation setting to a "direct sound emission system." While the paper presents high-quality datasets, the artificial conditions of the experiment, particularly the fluidic movement caused by direct sound stimulation, raise concerns about the reliability of the findings. Moreover, many of the findings in this manuscript are not novel, except for the use of audible sound frequencies. Lastly, the paper largely lacks sufficient discussion of the biological implications and follow-up questions stemming from these results.

1. The authors used a "direct sound emission system with a vibrational transducer to generate waves directly in the culture medium." However, at the frequency ranges used in this study (440 Hz and 14 kHz), significant fluid movements would likely be generated by the transducer. In this setup, it is unclear whether the cellular responses detected were due to the sound waves themselves or the fluid movements caused by the vibrations.
2. It seems unlikely that 14 kHz sound waves would fail to induce any flow or movement in the medium. Even high-frequency vibrations (or ultrasound) can cause localized disturbances in the fluid, though the type and scale of movement may differ from those generated by lower frequencies. Additionally, culture mediums are typically oxygenated from the surface, and static cultures can often maintain adequate oxygenation. Moreover, hypoxia-related genes can be induced by various types of cellular stress, not just oxygen deprivation, but also oxidative stress, mechanical stress, and metabolic changes.
3. The biological implications of comparing different waveforms remain unclear, particularly if this study was intended to generate new biological insights. Similarly, the analysis of early-response genes lacks clarity in terms of what novel information the reader is expected to gain.
4. It is interesting that some genes (such as *Ptgs2* and *Ctgf*) appear to be independent of cell confluency, while others (*Cyp11a1* and *Selp*) are highly dependent. The authors' demonstration of FAK-dependent mechanical sensing and the associated increase in cell area due to sound stimulation is noteworthy.
5. While the differences in cellular responses among different cell types are intriguing, the authors do not provide a clear hypothesis to explain the varied sensitivity. Are there conserved components in the relatively sensitive cell types compared to others? While the data is interesting, the manuscript lacks deeper interpretation, follow-up questions, or additional studies that could explore these findings further.
6. There have been multiple previous studies reporting the suppressive effects of low-intensity pulsed ultrasound on adipocyte differentiation. Although this study used audible frequencies, the difference in frequency alone does not sufficiently differentiate the current findings from existing literature. The authors should clarify how using audible frequencies provides new insights or advantages over what has already been established with ultrasound, particularly in terms of biological mechanisms or potential applications.

Reviewer #2

(Remarks to the Author)

This manuscript describes the acoustic effect on animal cells under well-established system. The objective and experimental design are very sound. I have couple of comments before publication.

1. Time and power: First, I am wondering the rationale to determine the sampling time like 2 and 24 h after sound treatment at 100 Pa. The timing and sound quality are critical for RNA-sequencing. I'd like to propose to evaluate the DEGs such as *Ptgs2*, *cox-2*, and *Ctgf* on early time points such as 10 min, 30 min, and 1 h. I believe the animal cell should respond more rapidly than we expect. Secondly, I am also questionable the 100 Pa. Is it biological relevant sound quality? Please explain more why authors chose the Pa. For sound quality, normally Hz and Decibel are broadly used. For the general audience, it is better to use general terms, I guess. Additionally, it needs to give more information why they used the murine C2C12 myoblast.
2. Sound transmission: I also have a question on the system. In current system, is the sound directly targeted into animal cells? If yes, how it can be survived the animal cell for 2 and 24 h without lid (cover for Petri-dish)? How does the authors prohibit the dehydration of the solution in the dish or well? I afraid the maintenance of animal cell without any side effect to evaluate the sole effect of acoustic treatment.
3. Discussion session: I think that these results are very intriguing for scientists and general audiences. Authors need to address the significance and meaning of this finding more in discussion session. For instance, adipocyte differentiation can be involved in many biological processes. How can you link the data to RNA-seq? What are the limitations of this study? What is the future perspective?

Reviewer #3

(Remarks to the Author)

This article investigated the effect of the acoustic stimulation on gene expression profiles and adipocyte differentiation in cultured mammalian cells. The authors performed a comprehensive analysis of gene expression profiles at different conditions and investigated the mechanisms. It is an interesting work, providing new insights into the biological responses of sound as a mechanical stimulus. I have a few questions and suggestions.

1. It is suggested to include specific details about the sound, such as the frequency and pressure level in the abstract.
2. Line 40-41, the sentence "Forces ranging from ..." should have literature to support.
3. Line 140-142. The claim that about 40% of sound waves can transmit from air into medium can not be true. Since there is a huge acoustic impedance mismatch between air and water. At the interface of air and water, there is almost total reflection. Less than 1% of the sound waves can transmit from air into water.
4. Can the author show some acoustic simulation results, in stead of just showing the dissected image in extended Data Fig.6?
5. It is suggested to provide a more in-depth discussion with other studies in which acoustic stimulation was applied.
6. What are the potential applications of these discoveries?

Version 1:

Reviewer comments:

Reviewer #1

(Remarks to the Author)

Comment #1.

1. The amplitude of a 440 Hz wave is about 5.92 times larger than that of a 14 kHz wave, not 32 times as initially stated. Please double-check your calculations.
2. The generation of a shear wave in a liquid involves factors beyond acoustic pressure, such as acoustic attenuation, which acts oppositely to amplitude differences. Therefore, the actual difference in effect could be even smaller than approximately 6-fold.
3. Nonetheless, despite the reduced amplitude difference, the distinct biological effects observed at different frequencies may still indicate differential gene activation mechanisms. It is important to further investigate whether these effects are primarily due to direct acoustic influence or if fluid movement within the medium plays a significant role.

Comment #2.

1. The acoustic radiation force from the 14 kHz sound wave could be significantly higher than that from the 440 Hz wave, given the same intensity. This variance might also contribute to hypoxia-related gene expression predominantly at 14 kHz. However, the manuscript lacks experimental validation regarding the claim of fluid movement. The reviewer strongly recommends conducting additional experiments to validate this aspect, as understanding the origin of differential gene expression is central to the core findings of this manuscript.

Reviewer #2

(Remarks to the Author)

I have no more comments. Authors gave appropriate responses from my questions. It is very interesting results.

Reviewer #3

(Remarks to the Author)

All my questions were addressed.

Version 2:

Reviewer comments:

Reviewer #1

(Remarks to the Author)

I find the manuscript acceptable for publication in its current form and have no further comments.

Reviewer #2

(Remarks to the Author)

No more comments.

Response to Referees:

Red: point-to-point responses to the referees' comments

Blue: citations of revised manuscript related to the comment

Reviewer #1:

In this manuscript, Kumenta et al. investigated cellular responses to audible sound using RNAseq and qPCR methods. They further explored the underlying sound-sensitive mechanisms and demonstrated the suppressive effects of acoustic stimulation on adipose differentiation. This study is an extension of their earlier paper published in 2018, where they examined cellular responses to "indirect audio sound" generated from a speaker. In this study, they modified their sound generation setting to a "direct sound emission system." While the paper presents high-quality datasets, the artificial conditions of the experiment, particularly the fluidic movement caused by direct sound stimulation, raise concerns about the reliability of the findings. Moreover, many of the findings in this manuscript are not novel, except for the use of audible sound frequencies. Lastly, the paper largely lacks sufficient discussion of the biological implications and follow-up questions stemming from these results.

1. The authors used a "direct sound emission system with a vibrational transducer to generate waves directly in the culture medium." However, at the frequency ranges used in this study (440 Hz and 14 kHz), significant fluid movements would likely be generated by the transducer. In this setup, it is unclear whether the cellular responses detected were due to the sound waves themselves or the fluid movements caused by the vibrations.

Thank you for pointing these important issues that need to be more clearly discussed in this manuscript. We appreciate this comment very much and carefully revised the manuscript.

Sound provides compressional mechanical wave that acts vertically to adherent cells, whereas fluid movement is expected to add shear stress in our experimental system. As mentioned in your comment, generation of acoustic wave in water inescapably accompanies fluid movement, that makes it impossible to experimentally assess the sole effect of compressional wave free of shear stress. Although it is difficult to obtain direct experimental proof, following findings strongly suggest that many of the cellular responses described in this manuscript were induced by the compressional sound waves:

When comparing acoustic waves with the same intensity, the amplitude of particle displacement of sine wave is inversely proportional to the sound frequency. Thus, under the same 100 Pa output, 400 Hz sound must have 32 times larger amplitude than 14 kHz, and generate much larger shear stress to the cells. Therefore, cell responses caused by the share stress are expected to be much higher in 400 Hz than 14 kHz. The correlation analyses of 440 Hz and 14 kHz gene responses presented in Fig.1f revealed dispersed distribution of the two. Further additional analyses revealed that almost equal amount of genes showed stronger response to 440 Hz (54.5%) and 14 kHz (45.5%). Histogram of the differences of the absolute 440 Hz response value from that of 14 kHz showed normal distribution with the peak value of 0.06. Therefore both 440 Hz and 14 kHz induced similar levels of unique gene responses, implying

acoustic wave as the major cause of these gene responses.

There may be some genes dominantly responded to the shear stress, rather than sound. We carefully revised the manuscript as follows to clearly tell that stronger effect of the shear stress may be included in the lower frequencies. Further studies comparing different frequency and output intensity will be useful to precisely estimate the effect of the fluid movements.

(page14-15, line248-266) Generation of acoustic wave in water invariably accompanies fluid movement. Although it is impossible to experimentally assess the sole effect of the compressional wave free of shear stress caused by the flow, comparison of the cell responses for 440 Hz and 14 kHz sound waves strongly suggests that many of the cellular responses observed in this study were induced by the compressional waves. When comparing acoustic waves with the same intensity, the amplitude of particle displacement of sine wave is inversely proportional to the sound frequency. Thus, under the same 100 Pa output, 440 Hz sound wave will have 32-fold larger amplitude than that of 14 kHz sound wave and generate much larger shear stress for the cells. Therefore, cell responses caused by the share stress are expected to be much higher at 440 Hz than those at 14 kHz. The correlation analyses of gene responses in response to 440 Hz and 14 kHz sound waves revealed dispersed distribution (Fig. 1f), with approximately half of genes (45.5%) showing stronger response to 14 kHz stimulation than to 440 Hz stimulation (Supplementary Fig. 10a). Histogram showing the differences in the absolute response at 440 Hz from that at 14 kHz revealed normal distribution with the peak value of 0.06 (Supplementary Fig. 10b). Therefore both 440 Hz and 14 kHz stimulation induced similar levels of unique gene responses, implying that compressional wave is the major cause of these gene responses. Notably, larger effects of the fluid movement may be included in the acoustic stimulation at lower frequencies in general. Further studies comparing different frequencies and output intensities will be useful to precisely estimate the effect of the fluid movements and elucidate the differences in results from those of compressional wave.

2. It seems unlikely that 14 kHz sound waves would fail to induce any flow or movement in the medium. Even high-frequency vibrations (or ultrasound) can cause localized disturbances in the fluid, though the type and scale of movement may differ from those generated by lower frequencies. Additionally, culture mediums are typically oxygenated from the surface, and static cultures can often maintain adequate oxygenation. Moreover, hypoxia-related genes can be induced by various types of cellular stress, not just oxygen deprivation, but also oxidative stress, mechanical stress, and metabolic changes.

We agree to the comment about the fluid movement. We revised the manuscript to tell the flow of the medium at 14 kHz stimulation as follows:

(page6, line86-89) the 440 Hz vibration caused convective mixing of the medium, while the 14 kHz high-frequency vibration induced much less flow, because the amplitude of particle displacement of sine wave is inversely proportional to the frequency.

In our system 80% of the surface of the culture medium was covered by the diaphragm. Although hypoxia-related genes could be induced by several types of stresses as you mentioned, we think it is more likely to be induced by the hypoxic condition of the medium, because of the attachment of the diaphragm. The fact that hypoxia-related response was only observed in 14 kHz stimulation which generates minimum fluid

movement also supports this interpretation. These interpretation was only described in the Supplementary Fig.3 legend before. Now we revised the manuscript to clearly tell this interpretation in the main text.

(page6, line86-93) This may be primarily due to the different fluid actions of the culture medium; the 440 Hz vibration caused convective mixing of the medium, while the 14 kHz high-frequency vibration induced much less flow, because the amplitude of particle displacement of sine wave is inversely proportional to the frequency. In our experimental system, a diaphragm covering 80% of the dish area was attached to the culture medium to efficiently generate acoustic waves. Taken together with the fluid action, 14 kHz stimulation generated a certain extent of hypoxia in the culture medium; as many of the 14 kHz responsive genes were found to be involved in the cellular response to hypoxia (Supplementary Fig. 3a).

3. The biological implications of comparing different waveforms remain unclear, particularly if this study was intended to generate new biological insights. Similarly, the analysis of early-response genes lacks clarity in terms of what novel information the reader is expected to gain.

Thank you for the comment. We performed data analysis to evaluate correlation coefficient between different sound patterns and revised the manuscript to include following discussion related to your suggestions.

(page15-16, line267-273) The effects of the different waveforms may be linked to different frequencies. Triangular and square waves contained more high-frequency overtone series than the pure sine wave (Supplementary Fig. 4); therefore, cell responses to triangular or square waves may include the effect of high-frequency waves. Comparison of the gene responses to 440 Hz and 14 kHz sound waves revealed higher correlation for square and triangular waves than sine wave, with the correlation coefficients of 0.76, 0.74, and 0.68 for square, triangular, and sine waves, respectively against 14 kHz, supporting this interpretation.

(page14, line237-244 All three sound factors (frequency, intensity, and waveform) affected gene response patterns, albeit in slightly different manners. Comparisons of 440 Hz and 14 kHz sound effects indicated that the early gene responses were similar, whereas frequency-specific responses were more prominent later (Fig. 1c–f). The effects on cell morphology within an hour were also similar among different frequencies (Fig. 3c), suggesting a cytoskeleton-linked shared sound response mechanism for the early response and unknown mechanisms that trigger frequency-specific cell responses in longer timeframes. Early response genes were more enriched in those involved in cell migration and adhesion (Fig. 1g), further supporting this idea.

4. It is interesting that some genes (such as *Ptgs2* and *Ctgf*) appear to be independent of cell confluency, while others (*Cyp1a1* and *Selp*) are highly dependent. The authors' demonstration of FAK-dependent mechanical sensing and the associated increase in cell area due to sound stimulation is noteworthy.

We revealed a FAK-dependent acoustic recognition mechanism that leads to a subset of early gene responses such as *Ptgs2* and *Ctgf*. These gene responses are shown to be independent of cell confluency (Fig. 2f). In general FAK-dependent signal transduction has been reported to be sensitive for cell density, since FAK is mainly involved in cell-matrix communication in a sparse condition, which is replace to cell-cell communication in a highly confluent situation. Our findings imply a novel form of FAK signal

transduction induced by acoustic stimulation which is independent of cell confluency. We revised the manuscript to include these implications in discussion section.

(page16, line280-284) In general FAK-dependent mechanical sensing is dependent on cell confluence, as FAK is mainly involved in cell-matrix communication in a sparse condition, which is replaced to cell-cell communication at high confluence[31]. FAK-dependent gene responses such as Ptgs2 and Ctgf under the condition of low as well high confluence (Fig. 2f) implies a novel form of FAK-dependent signal transduction triggered by acoustic stimulation.

[31] Zhang, L., Bewick, M. & Lafrenie, R. M. Role of Raf-1 and FAK in cell density-dependent regulation of integrin-dependent activation of MAP kinase. *Carcinogenesis* 23, 1251-8 (2002).

5. While the differences in cellular responses among different cell types are intriguing, the authors do not provide a clear hypothesis to explain the varied sensitivity. Are there conserved components in the relatively sensitive cell types compared to others? While the data is interesting, the manuscript lacks deeper interpretation, follow-up questions, or additional studies that could explore these findings further.

Thank you for the constructive comment. We added following discussion in the manuscript to provide insights into the cell type-dependency in acoustic response.

(page17-18, line302-311) Differences in sensitivity to acoustic stimulation likely reflect structural and functional heterogeneity of the focal adhesion in different cell types. The size, number, and molecular components of the focal adhesions vary significantly and are closely related to the cell motility and adhesion properties[38]. Adhesive stromal and its derivative cells including fibroblasts, myoblasts, osteoblasts, and adipocytes were highly sensitive to acoustic stimulation (Fig 4b), possibly due to their highly adhesive and motile nature that is essential to develop active focal adhesions. In contrast, epithelial and neuroblastoma cells, which are less mobile or less adhesive, were relatively insensitive. Although this focal adhesion-dependent pathway preferentially acts in stromal cell lines, other sound perception pathways may be active in other cell types. Further studies comparing the global gene expression patterns of different cell lines in response to different sound patterns unveil mechanisms underlying cell responses to sound.

[38] Zamir, E. & Geiger, B. Molecular complexity and dynamics of cell-matrix adhesions. *J Cell Sci* 114, 3583-90 (2001).

6. There have been multiple previous studies reporting the suppressive effects of low-intensity pulsed ultrasound on adipocyte differentiation. Although this study used audible frequencies, the difference in frequency alone does not sufficiently differentiate the current findings from existing literature. The authors should clarify how using audible frequencies provides new insights or advantages over what has already been established with ultrasound, particularly in terms of biological mechanisms or potential applications.

Thank you for pointing this out. We believe the signal transduction pathway involved in the response to acoustic stimulation to be different from that of ultrasound, and revised the manuscript to include following interpretation in the discussion section.

(page16-17, line289-296) Several studies have reported the effects of LIPUS in suppressing adipocyte differentiation. In response to LIPUS transmitted at different MHz frequencies, three cellular pathways are reportedly activated: ERK signalling through Rho-associated kinase or insulin receptor signaling[35,36], YAP nuclear translocation[36], and

histone deacetylase 1[37], all resulting in the suppression of adipocyte differentiation. The focal adhesion-mediated pathway revealed in this study is different from the pathway activated in response to LIPUS, which is expected considering an approximately 10^4 -fold difference in the frequency between LIPUS and audible sound sources used in this study.

(page17, line298-301) Considering the infinite sound pattern with both temporal and compositional variations, acoustic stimulation may induce diverse cellular responses and therefore, is an intriguing tool for cell manipulation such as living tissue engineering, regenerative medication, artificial tissue culture and related biotechnology industry.

(page18, line312-316) Sound exists universally in the material world; thus, life forms are exposed to this physical energy from their inception. It is, therefore, not surprising that living organisms develop systems to utilise sound as an environmental stimulus and optimise activities at the cellular level. Our findings reveal the nature of cells that sense acoustic information to orchestrate inner activities and mark the beginning of research focusing on the fundamental relationship between life and sound.

[35] Kusuyama, J. et al. Low intensity pulsed ultrasound (LIPUS) influences the multilineage differentiation of mesenchymal stem and progenitor cell lines through ROCK-Cot/Tpl2-MEK-ERK signaling pathway. *J Biol Chem* 289, 10330-10344 (2014).

[36] Nishida, T. et al. Suppression of adipocyte differentiation by low-intensity pulsed ultrasound via inhibition of insulin signaling and promotion of CCN family protein 2. *J Cell Biochem* 121, 4724-4740 (2020).

[37] Xu, T. et al. Low-intensity pulsed ultrasound inhibits adipogenic differentiation via HDAC1 signalling in rat visceral preadipocytes. *Adipocyte* 8, 292-303 (2019).

Reviewer #2:

This manuscript describes the acoustic effect on animal cells under well-established system. The objective and experimental design are very sound. I have couple of comments before publication.

1. Time and power: First, I am wondering the rationale to determine the sampling time like 2 and 24 h after sound treatment at 100 Pa. The timing and sound quality are critical for RNA-sequencing. I'd like to propose to evaluate the DEGs such as *Ptgs2*, *cox-2*, and *Ctgf* on early time points such as 10 min, 30 min, and 1 h. I believe the animal cell should respond more rapidly than we expect. Secondly, I am also questionable the 100 Pa. Is it biological relevant sound quality? Please explain more why authors chose the Pa. For sound quality, normally Hz and Decibel are broadly used. For the general audience, it is better to use general terms, I guess. Additionally, it needs to give more information why they used the murine C2C12 myoblast.

Thank you for pointing these out. We performed additional experiments to evaluate gene responses in shorter time points (10m, 30m, 1h, and 2h) for two sound-sensitive marker genes (*Ptgs2* and *Ctgf*), and found that statistically significant responses were observed at around 1~2h after the acoustic stimulation. Although there may be some immediate response genes within 2h, we concluded that 2h seems to be a reasonable timeframe to detect early gene responses in our RNA-seq screening assay. We added this additional experimental result in Supplementary Fig. 9 and revised the manuscript as follows:

(page14, line244-247) Evaluation of the gene responses at shorter time points revealed that 2 h was a reasonable timeframe to detect the expression of early response genes, as two marker genes *Ptgs2* and *Ctgf* showed statistically significant response in 1–2 h after the acoustic stimulation (Supplementary Fig. 9).

We selected 100 Pa intensity for several reasons: 1) sound pressure in our body is estimated to be up to several kPa by our simulation analysis and other reports, as described around line 64. Therefore 100 Pa is supposed to be a physiological range of sound intensity. 2) 100 Pa was almost the maximum intensity in our experimental setup, especially considering the effect of heat from the transducer, as described in Supplementary Fig.3. We revised the manuscript to include more detailed description about the 100 Pa intensity.

(page5, line67-70) Using this system, a variety of acoustic waves can directly be transmitted to cultured cells at the maximum intensity of approximately 100 Pa considering the effect of heat (Supplementary Fig. 2f), which is within the physiological range of sound pressure in living tissues.

The reason for using pascal (Pa) for sound intensity is to avoid confusion caused by the differences between air and water. Because sound was directly generated in culture medium in our system, we measured sound intensity in water. Although decibel (dB) is often used for sound in the air, differences in the reference intensity for dB in air and in water (air: 0 dB = 20 μ Pa, water: 0 dB = 1 μ Pa), and acoustic impedance (air: 413 NS/m³, water: 1.48M NS/m³) make it difficult to simply compare dB in air and water. Therefore we chose to describe sound intensity in Pa throughout the manuscript.

In a previous study using indirect sound emission system[1], we found C2C12 cells to be one of the most sound-sensitive cells among 4 different cell lines. We revised the manuscript to include the reason for choosing C2C12 cells.

(page5, line71-72) The murine C2C12 myoblast cell line was selected to investigate acoustic responses since it showed significant gene responses in our previous study[12].

[12] Kumeta, M., Takahashi, D., Takeyasu, K. & Yoshimura, S. H. Cell type-specific suppression of mechanosensitive genes by audible sound stimulation. *PLoS One* 13, e0188764 (2018).

2. Sound transmission: I also have a question on the system. In current system, is the sound directly targeted into animal cells? If yes, how it can be survived the animal cell for 2 and 24 h without lid (cover for Petri-dish)? How does the authors prohibit the dehydration of the solution in the dish or well? I afraid the maintenance of animal cell without any side effect to evaluate the sole effect of acoustic treatment.

Thank you for the comment. The diaphragm used in our system was designed to cover 80% of the dish surface. Therefore we believe the environment in the dish is properly maintained because the dish is not open to the air as the diaphragm is expected to act as a lid to avoid too much dehydration. On the other hand, it may induce oxidation of the medium by limiting the circulation of the CO₂ gas, especially in 14 kHz sound condition as it generate minimal fluid movement compared to 440 Hz and white noise. We revised the manuscript to describe the experimental design in detail and discuss potential effect of the oxidation as follows:

(page6, line84-93) a set of genes significantly upregulated by the 14 kHz sound was downregulated by the 440 Hz

sound (indicated in red in Fig. 1f). This may be primarily due to the different fluid actions of the culture medium; the 440 Hz vibration caused convective mixing of the medium, while the 14 kHz high-frequency vibration induced much less flow, because the amplitude of particle displacement of sine wave is inversely proportional to the frequency. In our experimental system, a diaphragm covering 80% of the dish area was attached to the culture medium to efficiently generate acoustic waves. Taken together with the fluid action, 14 kHz stimulation generated a certain extent of hypoxia in the culture medium as many of the 14 kHz responsive genes were found to be involved in the cellular response to hypoxia (Supplementary Fig. 3a).

3. Discussion session: I think that these results are very intriguing for scientists and general audiences. Authors need to address the significance and meaning of this finding more in discussion session. For instance, adipocyte differentiation can be involved in many biological processes. How can you link the data to RNA-seq? What are the limitations of this study? What is the future perspective?

Thank you for this constructive comment. We revised the manuscript to include 1) Unique feature of the effect of sound on adipocyte differentiation process, especially in comparison to the known effect of ultrasound, 2) Technical difficulty in separating the effect of fluid movement from compressional acoustic wave, as a current limitation of this study, 3) Biological significance and potential application of the findings presented in this study.

(page16-17, line288-298) Low-intensity pulsed ultrasound (LIPUS) at several MHz frequencies also induces Ptg2 and Ctgf expression in bone cells, such as osteoblasts and chondrocytes[33,34]. Several studies have reported the effects of LIPUS in suppressing adipocyte differentiation. In response to LIPUS transmitted at different MHz frequencies, three cellular pathways are reportedly activated: ERK signalling through Rho-associated kinase or insulin receptor signaling[35,36], YAP nuclear translocation[36], and histone deacetylase 1[37], all resulting in the suppression of adipocyte differentiation. The focal adhesion-mediated pathway revealed in this study is different from the pathway activated in response to LIPUS, which is expected considering an approximately 10^4 -fold difference in the frequency between LIPUS and audible sound sources used in this study. Several genes showed unique and characteristic responses to different sound stimulations (Fig. 1, Supplementary Fig. 3), suggesting a unique feature of acoustic stimulation that differs from other mechanical stimuli.

(page14-15, line248-266) Generation of acoustic wave in water invariably accompanies fluid movement. Although it is impossible to experimentally assess the sole effect of the compressional wave free of shear stress caused by the flow, comparison of the cell responses for 440 Hz and 14 kHz sound waves strongly suggests that many of the cellular responses observed in this study were induced by the compressional waves. When comparing acoustic waves with the same intensity, the amplitude of particle displacement of sine wave is inversely proportional to the sound frequency. Thus, under the same 100 Pa output, 440 Hz sound wave will have 32-fold larger amplitude than that of 14 kHz sound wave and generate much larger shear stress for the cells. Therefore, cell responses caused by the shear stress are expected to be much higher at 440 Hz than those at 14 kHz. The correlation analyses of gene responses in response to 440 Hz and 14 kHz sound waves revealed dispersed distribution (Fig. 1f), with approximately half of genes (45.5%) showing stronger response to 14 kHz stimulation than to 440 Hz stimulation (Supplementary Fig. 10a). Histogram showing the differences in the absolute response at 440 Hz from that at 14 kHz revealed normal

distribution with the peak value of 0.06 (Supplementary Fig. 10b). Therefore both 440 Hz and 14 kHz stimulation induced similar levels of unique gene responses, implying that compressional wave is the major cause of these gene responses. Notably, larger effects of the fluid movement may be included in the acoustic stimulation at lower frequencies in general. Further studies comparing different frequencies and output intensities will be useful to precisely estimate the effect of the fluid movements and elucidate the differences in results from those of compressional wave.

(page17, line298-301) Considering the infinite sound pattern with both temporal and compositional variations, acoustic stimulation may induce diverse cellular responses and therefore, is an intriguing tool for cell manipulation such as living tissue engineering, regenerative medication, artificial tissue culture and related biotechnology industry.

(page18, line312-316) Sound exists universally in the material world; thus, life forms are exposed to this physical energy from their inception. It is, therefore, not surprising that living organisms develop systems to utilise sound as an environmental stimulus and optimise activities at the cellular level. Our findings reveal the nature of cells that sense acoustic information to orchestrate inner activities and mark the beginning of research focusing on the fundamental relationship between life and sound.

[35] Kusuyama, J. et al. Low intensity pulsed ultrasound (LIPUS) influences the multilineage differentiation of mesenchymal stem and progenitor cell lines through ROCK-Cot/Tpl2-MEK-ERK signaling pathway. *J Biol Chem* 289, 10330-10344 (2014).

[36] Nishida, T. et al. Suppression of adipocyte differentiation by low-intensity pulsed ultrasound via inhibition of insulin signaling and promotion of CCN family protein 2. *J Cell Biochem* 121, 4724-4740 (2020).

[37] Xu, T. et al. Low-intensity pulsed ultrasound inhibits adipogenic differentiation via HDAC1 signalling in rat visceral preadipocytes. *Adipocyte* 8, 292-303 (2019).

Reviewer #3:

This article investigated the effect of the acoustic stimulation on gene expression profiles and adipocyte differentiation in cultured mammalian cells. The authors performed a comprehensive analysis of gene expression profiles at different conditions and investigated the mechanisms. It is an interesting work, providing new insights into the biological responses of sound as a mechanical stimulus. I have a few questions and suggestions.

1. It is suggested to include specific details about the sound, such as the frequency and pressure level in the abstract.

Thank you for the comment. We revised the abstract as follows:

(page2, line14-16) Here, we established a direct sound emission system with a vibrational transducer, and acoustic waves at frequency 440 Hz, 14 kHz, and white noise were transmitted to the murine C2C12 myoblasts at 100 Pa intensity.

2. Line 40-41, the sentence "Forces ranging from ..." should have literature to support.

We appreciate your suggestion. Because this is the general knowledge presented in the beginning of

introduction, we decided to add the following review paper as a reference here to support the sentence:
[1] Iskratsch, T., Wolfenson, H. & Sheetz, M. P. Appreciating force and shape-the rise of mechanotransduction in cell biology. *Nat Rev Mol Cell Biol* 15, 825-33 (2014).

3. Line 140-142. The claim that about 40% of sound waves can transmit from air into medium can not be true. Since there is a huge acoustic impedance mismatch between air and water. At the interface of air and water, there is almost total reflection. Less than 1% of the sound waves can transmit from air into water.

Thank you for pointing this out. We found this sentence really misleading. Here we did not aim to describe sound transmission efficiency from air to the medium (less than 1%, as you pointed), but aimed to explain the transmission pathways of sound detected in the medium. We performed simulation analyses under several different conditions shown in Supplementary Fig.6 and revealed that, among the total sound pressure detected in water, about 40% is supposed to transmit from air to the medium (① in the Fig), 13% by the vibration of the dish (②), and 47% by the reflection from the baseplate (③). We carefully revised the manuscript to clearly describe these results.

(page8-9, line137-140) To assess the sound transmission modes of the loudspeaker system, a sound-field simulation was performed with different combinations of water, dish, and baseplate. Comparison of the pressures observed at the detection point revealed revealing that among the total sound reaching the cells, the sound transmitted from the air to the medium only accounted for ~40% of the total sound reaching the cells (Supplementary Fig. 5).

4. Can the author show some acoustic simulation results, in stead of just showing the dissected image in extended Data Fig.6?

We considered carefully about the contents of the Supplementary Fig. 5 (previous Extended Data Fig.6), and concluded that there is no more useful information from the simulation analysis to present here. Because the frequency used for the simulation was very low compared to the object, sound intensity level remained quite uniformed within the system. There was no other processed data to show (frequency characteristics, for example, because only a single frequency was used here). Therefore we would like to leave this Figure as it was.

5. It is suggested to provide a more in-depth discussion with other studies in which acoustic stimulation was applied.

Thank you for the comment. We revised the discussion section to include unique feature of the effect of acoustic stimulation in inducing gene responses, influencing signal transduction, and affecting adipocyte differentiation process, especially in comparison to the known effect of ultrasound.

(page16-17, line288-301) Low-intensity pulsed ultrasound (LIPUS) at several MHz frequencies also induces Ptg2 and Ctgf expression in bone cells, such as osteoblasts and chondrocytes[33,34]. Several studies have reported the effects of LIPUS in suppressing adipocyte differentiation. In response to LIPUS transmitted at different MHz frequencies, three cellular pathways are reportedly activated: ERK signalling through Rho-associated kinase or

insulin receptor signaling[35,36], YAP nuclear translocation[36], and histone deacetylase 1[37], all resulting in the suppression of adipocyte differentiation. The focal adhesion-mediated pathway revealed in this study is different from the pathway activated in response to LIPUS, which is expected considering an approximately 10^4 -fold difference in the frequency between LIPUS and audible sound sources used in this study. Several genes showed unique and characteristic responses to different sound stimulations (Fig. 1, Supplementary Fig. 3), suggesting a unique feature of acoustic stimulation that differs from other mechanical stimuli. Considering the infinite sound pattern with both temporal and compositional variations, acoustic stimulation may induce diverse cellular responses and therefore, is an intriguing tool for cell manipulation such as living tissue engineering, regenerative medication, artificial tissue culture and related biotechnology industry.

[35] Kusuyama, J. et al. Low intensity pulsed ultrasound (LIPUS) influences the multilineage differentiation of mesenchymal stem and progenitor cell lines through ROCK-Cot/Tpl2-MEK-ERK signaling pathway. *J Biol Chem* 289, 10330-10344 (2014).

[36] Nishida, T. et al. Suppression of adipocyte differentiation by low-intensity pulsed ultrasound via inhibition of insulin signaling and promotion of CCN family protein 2. *J Cell Biochem* 121, 4724-4740 (2020).

[37] Xu, T. et al. Low-intensity pulsed ultrasound inhibits adipogenic differentiation via HDAC1 signalling in rat visceral preadipocytes. *Adipocyte* 8, 292-303 (2019).

6. What are the potential applications of these discoveries?

We appreciate your suggestion. We revised the manuscript to include biological significance and potential application of the findings presented in this study as follows:

(page17, line298-301) Considering the infinite sound pattern with both temporal and compositional variations, acoustic stimulation may induce diverse cellular responses and therefore, is an intriguing tool for cell manipulation such as living tissue engineering, regenerative medication, artificial tissue culture and related biotechnology industry.

(page18, line312-316) Sound exists universally in the material world; thus, life forms are exposed to this physical energy from their inception. It is, therefore, not surprising that living organisms develop systems to utilise sound as an environmental stimulus and optimise activities at the cellular level. Our findings reveal the nature of cells that sense acoustic information to orchestrate inner activities and mark the beginning of research focusing on the fundamental relationship between life and sound.

Response to Referees:

Red: point-to-point responses to the referees' comments

Blue: citations of revised manuscript related to the comment

Reviewer #1:

Comment #1.

1. The amplitude of a 440 Hz wave is about 5.92 times larger than that of a 14 kHz wave, not 32 times as initially stated. Please double-check your calculations.

Here we were describing the amplitude of particle displacement of sine wave.

[A: amplitude, ω : angular frequency, Z_0 : acoustic impedance, I: intensity]

$$v = \omega A \cos(\omega t)$$

$$v_0 = \omega A$$

$$Z_0 = \frac{p}{v}$$

$$\text{Therefore } p_0 = v_0 \times Z_0 = \omega A Z_0$$

$$I = p \times v = \frac{p^2}{Z_0} = \frac{p_{rms}^2}{Z_0} = \frac{\left(\frac{p_0}{\sqrt{2}}\right)^2}{Z_0} = \frac{p_0^2}{2 \times Z_0} = \frac{(\omega A)^2 \times Z_0}{2}$$

Therefore when under constant intensity (I), the amplitude of particle displacement of sine wave (A) is inversely proportional to the frequency (ω).

Although there are several parameters to explain differences of the dynamics of the medium in different frequencies, we decided to describe the amplitude of particle displacement of sine wave, because we utilized sine waves as the major form of acoustic stimulation in this manuscript.

2. The generation of a shear wave in a liquid involves factors beyond acoustic pressure, such as acoustic attenuation, which acts oppositely to amplitude differences. Therefore, the actual difference in effect could be even smaller than approximately 6-fold.

Acoustic attenuation in water can be calculated as follows.

[α : attenuation coefficient (dB/MHz*cm), d: distance (cm), f (MHz)]

$$\text{attenuation (dB)} = \alpha \times d \times f$$

Calculation using attenuation coefficient of water (0.0022) and actual distance between sound input source and cells in our experimental system (0.1 cm) revealed 9.68×10^{-6} and 3.08×10^{-8} (dB) attenuation for 440 Hz and 14 kHz sound, respectively. Therefore the effect of acoustic attenuation is negligibly minimal.

3. Nonetheless, despite the reduced amplitude difference, the distinct biological effects observed at different frequencies may still indicate differential gene activation mechanisms. It is important to further

investigate whether these effects are primarily due to direct acoustic influence or if fluid movement within the medium plays a significant role.

As we explained in the previous response, generation of acoustic wave in water inescapably accompanies fluid movement, that makes it impossible to experimentally assess the sole effect of acoustic wave free of shear stress. However, although it is difficult to obtain direct experimental proof, following results and findings suggest that many of the cellular responses described in this manuscript were induced by the acoustic waves:

If fluid movement plays a significant role in the gene responses, 440 Hz sound must show much higher response than that of 14 kHz, because of the stronger shear stress. However, the correlation analyses of 440 Hz and 14 kHz gene responses presented in Fig.1f revealed evenly dispersed distribution of the two. When the fold differences of genes listed in Fig 1d and 1e against 440 Hz and 14 kHz stimulations were compared, the number of genes which showed stronger response to 440 Hz was almost half (54.5%) of them. Histogram of the differences of absolute 440 Hz response value from that of 14 kHz showed normal distribution with the peak value of 0.06. Therefore both 440 Hz and 14 kHz induced similar levels of unique gene responses, implying acoustic wave as the major cause of these gene responses.

There may be some genes more affected by the shear stress, rather than sound. We carefully revised the manuscript as follows to clearly tell these interpretation, and revised the manuscript as follows:

(page14-15, line248-263) Generation of acoustic wave in water invariably accompanies fluid movement. Although it is impossible to experimentally assess the sole effect of the compressional wave free of shear stress caused by the flow, comparison of the cell responses for 440 Hz and 14 kHz sound waves strongly suggests that many of the cellular responses observed in this study were induced by the compressional waves. When comparing acoustic waves with the same intensity, the amplitude of particle displacement of sine wave is inversely proportional to the sound frequency. Thus, under the same 100 Pa output, 440 Hz sound wave will have 32-fold larger amplitude than that of 14 kHz sound wave and generate much larger shear stress for the cells. Therefore, cell responses caused by the shear stress are expected to be much higher at 440 Hz than those at 14 kHz. The correlation analyses of gene responses in response to 440 Hz and 14 kHz sound waves revealed dispersed distribution (Fig. 1f), with approximately half of genes (45.5%) showing stronger response to 14 kHz stimulation than to 440 Hz stimulation (Supplementary Fig. 10a). Histogram showing the differences in the absolute response at 440 Hz from that at 14 kHz revealed normal distribution with the peak value of 0.06 (Supplementary Fig. 10b). Therefore both 440 Hz and 14 kHz stimulation induced similar levels of unique gene responses, implying that compressional wave is the major cause of these gene responses.

(page15, line267-271) Notably, larger effects of the fluid movement may be included in the acoustic stimulation at lower frequencies in general. Further studies comparing different frequencies and output intensities will be useful to precisely estimate the effect of the fluid movements and elucidate the differences in results from those of compressional wave.

Comment #2.

1. The acoustic radiation force from the 14 kHz sound wave could be significantly higher than that from

the 440 Hz wave, given the same intensity. This variance might also contribute to hypoxia-related gene expression predominantly at 14 kHz. However, the manuscript lacks experimental validation regarding the claim of fluid movement. The reviewer strongly recommends conducting additional experiments to validate this aspect, as understanding the origin of differential gene expression is central to the core findings of this manuscript.

We understand your concern. However as discussed in the previous response, it is impossible to experimentally assess the effect of acoustic wave free from the fluid movement. Meanwhile, many researches focusing on the shear stress have identified characteristic gene responses. For example, 78 genes were found to respond laminar or turbulent shear stress in a global gene analysis using vascular endothelial cells [1] and 254 genes were annotated as “response to fluid shear stress (GO:0034405)” in the gene ontology database. We performed additional analysis to figure out the overlap between sound-sensitive and shear stress-sensitive genes, and found only 4 genes overlap with the global gene analysis and 3 genes with the gene ontology database. The P value for the enrichment of sound-sensitive genes in “response to fluid shear stress (GO:0034405)” was 0.0321, which was not ranked in the top 40 annotation categories. Taken all together, we concluded that the effect of the fluid shear stress was not significantly large, and cellular responses described in this manuscript are likely to be induced by the acoustic wave. We thank the reviewer to point these things out, as we could discuss the effect of fluid movement more in detail in the manuscript.

(page15, line263-267) This interpretation is also supported by an annotation analysis. Among 254 genes annotated as “response to fluid shear stress (GO:0034405)” in gene ontology database, only 3 genes overlap with sound-sensitive genes identified in this study (Fig. 1d, 1e). This annotation category was not found in the top 40 annotation clusters enriched in the sound-sensitive genes, suggesting a minimal influence of the fluid shear force in our acoustic experimental condition.

[1] Ohura N, Yamamoto K, Ichioka S, Sokabe T, Nakatsuka H, Baba A, Shibata M, Nakatsuka T, Harii K, Wada Y, Kohro T, Kodama T, Ando J. Global analysis of shear stress-responsive genes in vascular endothelial cells. *J Atheroscler Thromb.* 2003;10(5):304-13.

Reviewer #2:

I have no more comments. Authors gave appropriate responses from my questions. It is very interesting results.

Reviewer #3:

All my questions were addressed.